# FLAME: Fast Long-context Adaptive Memory for Event-based Vision

**Biswadeep Chakraborty**
Georgia Institute of Technology,
Mercuria Energy Trading
bchakraborty6@gatech.edu

**Saibal Mukhopadhyay**
School of Electrical and Computer Engineering
Georgia Institute of Technology

## Abstract

We propose Fast Long-context Adaptive Memory for Event (FLAME), a novel scalable architecture that combines neuro-inspired feature extraction with robust structured sequence modeling to efficiently process asynchronous and sparse event camera data. As a departure from conventional input encoding methods, FLAME presents Event Attention Layer, a novel feature extractor that leverages neuromorphic dynamics (Leaky Integrate-and-Fire (LIF)) to directly capture multi-timescale features from event streams. The feature extractor integrates with a structured state-space model with a novel Event-Aware HiPPO (EA-HiPPO) mechanism that dynamically adapts memory retention based on inter-event intervals to understand relationship across varying temporal scales and event sequences. A Normal Plus Low Rank (NPLR) decomposition reduces the computational complexity of state update from $\mathcal{O}(N^2)$ to $\mathcal{O}(Nr)$, where $N$ represents the dimension of the core state vector and $r$ is the rank of a low-rank component (with $r \ll N$). FLAME demonstrates state-of-the-art accuracy for event-by-event processing on complex event camera datasets.

## 1 Introduction

Neuromorphic cameras promise low operation power by generating asynchronous and sparse event data instead of dense pixel maps. Although spatial and temporal resolutions of event cameras are increasing [1, 2, 3], extracting features from long sequence of events remains challenging and computationally expensive [4, 5]. One class of event camera algorithms aggregate events over a period of time into a volumetric representation [6, 7, 8, 9]. They show higher accuracy but incur higher computational cost and longer decision latency. Alternatively, event-by-event processing methods maintains a spatiotemporal representation and updates that for every new event[10, 4, 11]. They require less computation and lower latency, but often achieve lower accuracy on complex datasets, simultaneously achieving low computational cost, event-by-event processing, and high accuracy for complex event camera data remains challenging.

Recent approaches for computationally efficient event camera processing primarily include methods based on Graph Neural Networks (GNNs), Transformers, and Spiking Neural Networks (SNNs). While GNNs and Transformers can model complex inter-event relationships to achieve high accuracy, their core operations (e.g., graph convolution and attention mechanisms) typically process events in aggregated batches or windows [12, 13, 14, 15, 16, 17]. This operational paradigm fundamentally limits their ability to perform continuous, low-latency event-by-event updates and often incurs high computational and memory demands, especially with large event volumes and long sequences [18, 19, 20]. Conversely, SNN-based methods inherently support event-by-event processing with promising computational efficiency. However, they often exhibit lower accuracy on complex tasks involving high spatial and temporal resolutions (i.e., longer event sequences). This performance gap for SNNs frequently stems from challenges in effectively training very deep architectures and the

39th Conference on Neural Information Processing Systems (NeurIPS 2025).

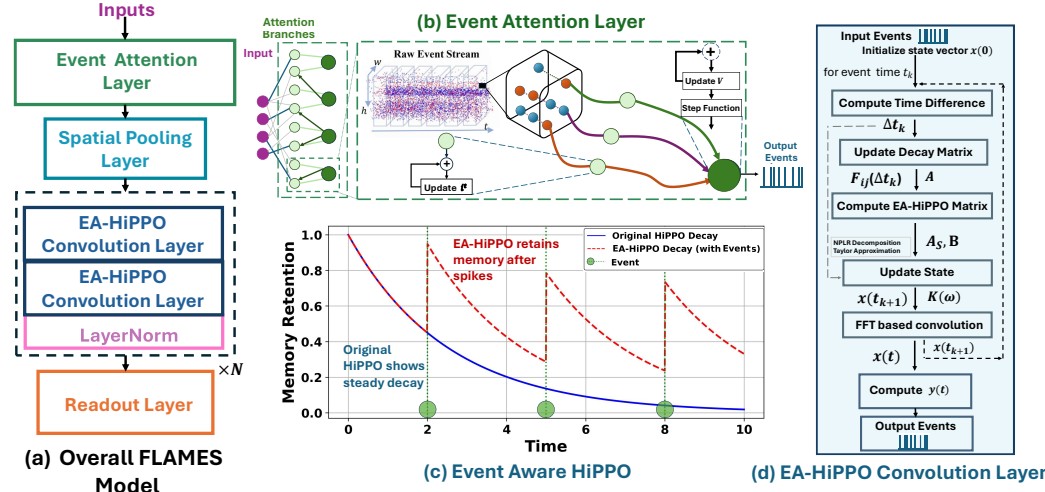

Figure 1: **Block diagram of the proposed FLAME architecture.** (**a**) Overall architecture, combining neuro-inspired feature extraction with efficient state-space modeling. (**b**) The **Event Attention Layer (EAL)** uses multi-branch Leaky Integrate-and-Fire (LIF) dynamics to extract multi-timescale temporal features from raw event streams. (**c**) The core **Event-Aware HiPPO (EA-HiPPO)** dynamically modulates memory retention based on event timing ($\Delta t$), retaining context better than standard HiPPO after sparse events. (**d**) The **EA-HiPPO Convolution Layer** achieves efficiency via asynchronous updates, Normal-Plus-Low-Rank (NPLR) decomposition, and FFT-based convolution.

inherent transient memory of basic spiking neuron models, which can impede the robust retention of contextual information over extended event sequences.

State Space Models (SSMs) have emerged as an effective paradigm for modeling complex sequences [21, 22]. They demonstrated good performance on event datasets but with event aggregation, and hence, high computational cost [23, 24]. Advanced SSMs use **HiPPO** (High-Order Polynomial Projection Operator) framework [25] to compress and represent an input's history through structured state-space dynamics. The SSMs built on HiPPO principles (e.g., S4 [21]) excel on dense, regularly sampled data such as images. However, the *HiPPO mechanism and subsequent SSM dynamics* inherently assume continuous or regularly-sampled inputs and require *initial encoding layers* to convert events into dense representations. Hence, they do not leverage the discrete, asynchronous, and sparse nature of event data for computational efficiency and event-by-event processing.

This paper presents **FLAME (Fast Long-context Adaptive Memory for Event-based Systems)**, a novel, computationally efficient, and scalable SSM-based framework for event-by-event processing of large-scale event-based vision data. The FLAME architecture makes following key contributions.

- **Novel Event-Driven Input Encoding with Neuromorphic Dynamics:** We propose the **Event Attention Layer**, which utilizes neuromorphic-inspired dynamics (such as Leaky Integrate-and-Fire mechanisms) for direct, multi-timescale feature extraction from raw asynchronous event streams. This approach overcomes limitations of conventional input encoders for SSMs when applied to event data, by inherently processing information event-by-event and preserving crucial temporal precision and sparsity without requiring dense, fixed-size input patches or hand-crafted features.

- **Event-Aware HiPPO (EA-HiPPO) for Adaptive Memory:** We present a novel adaptation of HiPPO principles where the state-space dynamics are made explicitly sensitive to the precise timing of discrete input events. Unlike standard HiPPO which assumes continuous inputs, EA-HiPPO dynamically modulates memory retention based on inter-event intervals. This allows it to effectively capture diverse temporal patterns in varying timescales and maintain context within sparse and asynchronous event streams.

- **Computationally Efficient SSM Framework for Events:** We apply Normal-Plus-Low-Rank (NPLR) decomposition within the EA-HiPPO core, which reduces state update com-

plexity from $\mathcal{O}(N^2)$ [21] to $\mathcal{O}(Nr)$. The reduced computational cost makes FLAME suitable for high-dimensional event streams.

Our theoretical analysis substantiates the FLAME framework, establishing bounds on memory capacity [26, 27] versus computational cost for these event-driven SSM representations. Our experimental results on multiple event camera datasets, namely, HAR-DVS, Celex-HAR, N-Caltech101, and CIFAR10-DVS, demonstrates that state-of-the-art accuracy with event-by-event processing but at a reduced computational cost (FLOPS/event) and processing latency. **Crucially, on high-resolution data like CeleX-HAR, FLAME achieves comparable accuracy ($\approx 72.2\%$) at $\approx 90\times$ lower compute (0.41 GFLOPs vs. 37.2 GFLOPs) compared to EventMamba, highlighting our efficiency focus.**

Table 1: Qualitative Comparison of FLAME with prior methods

| Model | Specific Type | Dynamic Memory Retention | Direct Event-by-Event Processing | Scalable Long-Context Memory | Low Computational Overhead | Fully Asynchronous Updates | Adaptability to Sparse Data |
|---|---|---|---|---|---|---|---|
| *Graph Neural Networks (GNN)* | | | | | | | |
| AEGNN [18] | Asynchronous GNN | ✗ | ✓ | ✗ | ✗ | ✓ | ✓ |
| EventGTS [28] | Tracking GNN | ✗ | ✗ | ✗ | ✗ | ✗ | ✓ |
| GEC-Net [29] | Classification GNN | ✗ | ✗ | ✗ | ✗ | ✗ | ✓ |
| *Transformers* | | | | | | | |
| EventNet (Alert-TF) [15] | Alert-driven TF | ✗ | *Partial* | *Partial* | ✗ | ✓ | ✓ |
| Spiking Transformer [14] | Spiking TF | *Partial* | ✓ | ✓ | *Partial* | ✓ | ✓ |
| RVT [30] | Recurrent Vision TF | ✓ | ✗ | *Partial* | ✗ | ✗ | ✗ |
| SpikeGPT [31] | Autoregressive Spiking TF | *Partial* | ✓ | *Partial* | *Partial* | ✓ | ✓ |
| EST [32] | Spatio-temporal TF | ✗ | ✗ | ✗ | ✗ | ✗ | ✗ |
| *Spiking Neural Networks (SNN)* | | | | | | | |
| DH-LIF [33] | Dual-threshold LIF SNN | ✗ | ✓ | ✗ | ✓ | ✓ | ✓ |
| HOTS [34] | Time-Surface (SNN-inspired) | ✓ | ✓ | ✗ | ✓ | ✓ | ✓ |
| SLAYER-SNN [35] | SNN (generic) | *Partial* | ✓ | ✗ | ✓ | ✓ | ✓ |
| LoCoSNN [36] | Low-complexity SNN | ✗ | ✓ | ✗ | ✓ | ✓ | ✓ |
| LIAF-Net [37] | LIF SNN w/ Attention | *Partial* | ✗ | ✗ | *Partial* | ✗ | ✓ |
| *State Space Models (SSM)* | | | | | | | |
| SpikingLMU [38] | Spiking LMU-TF (Hybrid) | ✓ | *Partial* | ✓ | ✗ | *Partial* | ✓ |
| BinaryS4D [39] | Binarized S4D (SSM) | ✗ | ✗ | ✓ | ✓ | ✗ | ✗ |
| S4 [21] (applied to events) | Structured SSM | ✗ | ✗ | ✓ | ✓ | ✗ | ✗ |
| EventMamba [40] | Hybrid CNN-SSM | *Partial* | ✗ | ✓ | ✗ | ✗ | ✗ |
| SSM-Event [23] | Structured SSM | ✓ | ✗ | ✓ | *Partial* | ✗ | ✗ |
| PRE-Mamba [24] | Mamba-based SSM | ✓ | ✗ | ✓ | *Partial* | ✗ | ✗ |
| FLAME (Ours) | **Event-Aware SSM** | ✓ | ✓ | ✓ | ✓ | ✓ | ✓ |

## 2   Related Works

**Event Processing with Graph Neural Networks** Graph Neural Networks (GNNs) model event data by constructing graphs over event windows or directly exploiting event sparsity [29, 28]. While capable of capturing complex spatio-temporal relationships (e.g., EventGTS [28], GEC-Net [29]), GNNs often process aggregated event batches, limiting low-latency updates and incurring costs for graph management with large event volumes. Though some aim for asynchronous updates (e.g., AEGNN [18]), GNNs generally lack the inherent, scalable long-context memory mechanisms that FLAME's SSM-based architecture provides without explicit graph construction.

**Event Processing with Transformers** Transformers [41], adapted for event data, range from operating on event volumes (RVT [30], EST [32]) to more event-driven variants like Alert-Transformer [15] (partially asynchronous) and Spiking Transformers [14, 31] (aiming for SNN efficiency). However, the core attention mechanism often remains computationally intensive for long event sequences or requires windowing strategies that can compromise fully asynchronous, low-latency updates [16, 42, 43]. FLAME utilizes efficient SSM recurrence, bypassing the quadratic complexity of full attention for modeling long temporal contexts.

**Spiking Neural Networks for Event Camera Processing** Spiking Neural Networks (SNNs) naturally suit event cameras' asynchronous, sparse nature, promising efficiency [4]. Standard training (SLAYER [35], surrogate gradients [44, 45]) has advanced SNNs (e.g., LoCoSNN [36]), which can process event-by-event. We choose LIF over simpler Integrate-and-Fire (IF) neurons because the leak is necessary to avoid unbounded build-up during long silent gaps and ensure training stability in deeper setups. However, traditional LIF neurons often struggle with extended temporal contexts due to rapid memory decay [46]. While enhancements like dendritic processing (DH-LIF [33]) exist,

FLAME uniquely integrates principles from such mechanisms in its initial Event Attention Layer with its novel EA-HiPPO SSM core for robust long-context modeling, a common challenge for standalone SNNs.

**State Space Models for Event Cameras** State Space Models (SSMs) like S4 [21] and Mamba [22], built on the HiPPO framework [25], excel at long-sequence modeling but are designed for dense, regularly sampled data, making them sub-optimal for raw, sparse, asynchronous event streams [47, 48, 49]. Applying them to events often requires initial aggregation (e.g., into grids [23] or voxels [40]), potentially losing fine temporal details. While recent works (SpikingLMU [38], BinaryS4D [39], SpikingSSMs [50], P-SpikeSSM [51]) explore bridging SSMs and event-driven principles, FLAME's Event-Aware HiPPO (EA-HiPPO) offers a distinct approach. It directly adapts core SSM state dynamics using measured inter-event time intervals, preserving HiPPO's long-context strengths while making memory evolution explicitly sensitive to event stream patterns.

**Adaptive Memory Mechanisms in Event-Driven Systems** Adaptive memory is crucial for event-driven systems [45, 52, 27]. Approaches include Liquid Time-Constant (LTC) Networks [48] (adapting neuron time-constants) or STDP-like local learning [53]. FLAME's EA-HiPPO presents a novel strategy: it achieves adaptivity by globally and explicitly modulating a structured SSM's memory dynamics based on the timing of incoming events [26, 54]. This direct modulation by event statistics makes EA-HiPPO highly effective for efficient long-context modeling in neuromorphic vision [55], temporal reasoning [56], and resource-constrained edge applications [57, 58, 59, 60].

## 3  Methods

FLAME introduces a novel neural network architecture engineered for the efficient processing of high-dimensional, asynchronous data from event cameras. It addresses the critical need for low computational overhead while effectively modeling complex spatio-temporal dynamics. The architecture (Figure 1) strategically employs neuromorphic-inspired, event-driven computation for its initial feature extraction and then integrates this with a powerful SSM core for robust temporal modeling. This design philosophy allows FLAME to directly process raw event data, harness its inherent sparsity, and overcome both the memory limitations of purely event-driven systems and the dense-input requirements of traditional SSMs. The data pipeline comprises: an **Event Attention Layer** for multi-timescale feature extraction from raw asynchronous events; a subsequent **Spatial Pooling Layer** to enhance computational efficiency by reducing dimensionality; the core **EA-HiPPO Convolution Layer** for modeling long-term sequential patterns through event-aware state-space dynamics; and finally, Layer Normalization and a **Readout Layer** for task-specific outputs.

**Input Representation:** FLAME directly processes input as a sequence of asynchronous events, the native output format of event cameras. Each event is typically a tuple $(x, y, t, p)$ denoting spatial coordinates, a precise timestamp, and polarity. This native, event-by-event processing is fundamental to FLAME's efficiency and its ability to preserve the rich temporal dynamics and sparsity of neuromorphic sensor data, as computation is triggered only by new information. Our goal with "event-by-event" is to ensure every update (feature extraction, pooling, and state updates) happens without batching or delay, not to imply that every stage is determined by a single input spike alone.

**Event Attention Layer** The initial stage of FLAME, the *Event Attention Layer*, is designed to effectively capture the complex and varying temporal dynamics present in asynchronous event streams from high-dimensional event cameras. This layer is essential for transforming the raw, sparse event data into a richer feature representation while managing its inherent irregularity. Inspired by the multi-timescale filtering capabilities of dendritic branches in biological neurons, this layer is constructed using LIF neurons, each augmented with multiple conceptual branches (Figure 1(b)). Each branch $d$ is characterized by a distinct learnable timing factor $\tau_d$, allowing it to act as a temporal filter sensitive to a specific timescale. This is particularly important for event data where information can be encoded across a wide spectrum of inter-event intervals.

The current $\mathbf{i}_d(t)$ in a conceptual branch $d$ evolves according to:

$$\frac{d\mathbf{i}_d(t)}{dt} = -\frac{1}{\tau_d}\mathbf{i}_d(t) + \sum_{j \in \mathcal{N}_d} \mathbf{w}_{dj} E_j(t), \qquad (1)$$

or in its discrete-time form often used for simulation with a step $\Delta t_{sim}$ (if applicable):

$$\mathbf{i}_d(t + \Delta t_{sim}) = e^{-\frac{\Delta t_{sim}}{\tau_d}} \mathbf{i}_d(t) + (1 - e^{-\frac{\Delta t_{sim}}{\tau_d}}) \sum_{j \in \mathcal{N}_d} \mathbf{w}_{dj} E_j(t), \tag{2}$$

where $E_j(t)$ represents an input event from source $j$ (binary, 1 if an event occurs at $t$, 0 otherwise), $\mathbf{w}_{dj}$ are the learnable weights connecting input $j$ to branch $d$, and $\mathcal{N}_d$ is the set of inputs to branch $d$. This multi-timescale filtering allows each neuron to effectively "attend" to features across diverse temporal windows within the sparse and irregular event stream, extracting meaningful patterns that might otherwise be lost.

These branch currents are then integrated at the neuron's soma $\tau_s \frac{dV(t)}{dt} = -V(t) + \sum_d \mathbf{g}_d \mathbf{i}_d(t)$, where $V(t)$ is the membrane potential, $\tau_s$ is the somatic membrane time constant, and $\mathbf{g}_d$ is the **learnable coupling strength** of branch $d$ to the soma. This late fusion mechanism, using independent branches and separate coupling weights $\mathbf{g}_d$, is explicitly designed to minimize cross-talk between fast- and slow-timescale responses, as the branches contribute separately to the final firing decision. An output occurs when $V(t)$ exceeds a threshold $V_{\text{th}}$, after which $V(t)$ is reset. The output of this layer is thus a new set of event trains, now enriched with multi-timescale temporal features, ready for further processing. The EAL acts as an effective denoising and temporal abstraction layer, suppressing isolated spikes while enhancing coherent spatio-temporal structures. This layer acts as an efficient front-end for raw event data, using LIF neuron dynamics for event generation which maintains sparsity.

**Spatial Pooling Layer** High-resolution event cameras (e.g., $1280 \times 800$) produce vast amounts of spatial data, posing computational challenges. To manage this efficiently, FLAME incorporates a *Spatial Pooling Layer* immediately after the Event Attention Layer. The intuition is to reduce spatial dimensionality *after* initial rich feature extraction but *before* more complex temporal modeling, preserving crucial spatio-temporal details that might be lost if pooling raw, unprocessed input. Given an input event map $I(x, y, t)$ from the Event Attention Layer, pooling (e.g., $k \times k$ max-pooling) produces $I_{\text{pooled}}(x', y', t) = \max_{(x,y) \in P(x',y')}\{I(x, y, t)\}$. This operation is performed per event timestamp using a local max operation, without using any voxel grid or frame-level aggregation, fully preserving temporal resolution. Ablations show that applying pooling *before* the EAL caused a consistent performance drop, justifying our chosen order. The output $I_{\text{pooled}}$ is typically flattened into $\mathbf{E}_{flat}(t) \in \{0, 1\}^M$ for the EA-HiPPO layer.. This operation, performed at each event time $t$, maintains precise event timing and significantly reduces the feature map size, critical for the computational tractability of subsequent layers. The output $E_{\text{pooled}}$ is typically flattened into $\mathbf{E}_{flat}(t) \in \{0, 1\}^M$ for the EA-HiPPO layer.

**EA-HiPPO Convolution Layer** The core of FLAME's capability to efficiently model long temporal contexts from event data resides in the *EA-HiPPO Convolution Layer*. This layer adapts State Space Model principles, particularly the HiPPO framework [25], to the sparse, event-driven nature of the processed event vector $\mathbf{E}_{flat}(t)$. While termed "convolution" due to its conceptual link to efficient parallel training modes of modern SSMs (e.g., S4), its inference is inherently recurrent and event-driven, aligning with low-compute goals.

A state vector $\mathbf{x}(t) \in \mathbb{R}^N$ captures historical context, evolving as $\dot{\mathbf{x}}(t) = \mathbf{A}_S(t)\mathbf{x}(t) + \mathbf{B}\mathbf{E}_{flat}(t)$. The novelty here is the time-varying adaptive state transition matrix $\mathbf{A}_S(t) \equiv \mathbf{A}_S(\Delta t) = \mathbf{A} \circ \mathbf{F}(\Delta t)$. Here, $\mathbf{A}$ is a base HiPPO matrix, and $\mathbf{F}(\Delta t)$ is a dynamic decay modulation matrix (with elements $F_{ij}(\Delta t) = e^{-\alpha_{ij}\Delta t}$ and learnable $\alpha_{ij} > 0$) that adapts to the inter-event interval $\Delta t$. This **Event-Aware HiPPO (EA-HiPPO)** design allows memory dynamics to respond to event sparsity: frequent events ($\Delta t \approx 0$) lead to $\mathbf{A}_S \approx \mathbf{A}$ (preserving memory), while sparse events (large $\Delta t$) allow faster decay of specific memory elements.

The matrix $\mathbf{A}$ is initialized using the HiPPO-LegS formulation, which guarantees that all its eigenvalues satisfy $\text{Re}(\lambda_i(\mathbf{A})) < 0$ (a Hurwitz matrix), ensuring the continuous-time system is asymptotically stable. This stability is preserved in the event-adaptive variant $\mathbf{A}_S(\Delta t)$ because the decay modulation matrix $\mathbf{F}(\Delta t) \in (0, 1]$ does not flip the sign of the eigenvalues.

This dynamic adaptation is crucial for asynchronous event streams. For event-driven discrete-time updates (interval $\Delta t_k$, input $\mathbf{E}_{event_{k+1}}$):

$$\mathbf{x}(t_{k+1}) = e^{\mathbf{A}_S(\Delta t_k)\Delta t_k}\mathbf{x}(t_k) + (\mathbf{A}_S(\Delta t_k))^{-1}(e^{\mathbf{A}_S(\Delta t_k)\Delta t_k} - \mathbf{I})\mathbf{B}\mathbf{E}_{event_{k+1}}. \tag{3}$$

The matrix exponential is efficiently approximated using a second-order Taylor expansion. This, along with Normal-Plus-Low-Rank (NPLR) decomposition of $\mathbf{A}$ (reducing complexity to $O(Nr)$), ensures low computational cost. Specifically, we use $N = 64$ and fix the rank to $r = 8$ in all experiments. The output state $\mathbf{x}(t)$ thus encodes long-term sequential patterns informed by precise event timings.

**Layer Normalization and Readout Layer** Following the EA-HiPPO Convolution Layer, Layer Normalization is applied to the state representation $\mathbf{x}(t)$ to stabilize learning from sparse and dynamic event-driven activations. The final *Readout Layer* then maps these learned temporal representations to task-specific predictions. It typically aggregates information from the state sequence $\mathbf{x}(t_1), \ldots, \mathbf{x}(t_L)$ (e.g., via temporal averaging of the last $K$ states to get $\mathbf{x}_{pooled}$) and passes this through a linear layer with an appropriate output activation (e.g., softmax for classification, $\mathbf{y}_{pred} = \text{Softmax}(\mathbf{W}_{out}\mathbf{x}_{pooled} + \mathbf{b}_{out})$). This provides an efficient mechanism to derive final outputs.

# 4 Theoretical Analysis

This section establishes theoretical guarantees for FLAME, our adaptive state-space framework. We focus on three key aspects: computational efficiency of state updates, the ability to model extended temporal contexts from event data, and overall system stability. Formal bounds are presented to analyze how FLAME's components ensure robust and efficient processing. ***Detailed proofs for all theorems and lemmas are provided in the Suppl. Material.***

## 4.1 Computational Complexity: Enabling Efficient Event Processing

**Lemma 1** (Computational Efficiency of Adaptive State Updates). *The state update for the EA-HiPPO dynamics, $\dot{\mathbf{x}}(t) = \mathbf{A}_S(\Delta t)\mathbf{x}(t) + \mathbf{B}\mathbf{E}_{flat}(t)$ (as defined in Sec. 3, where $\mathbf{E}_{flat}(t)$ is the flattened input event vector), with $\mathbf{x}(t) \in \mathbb{R}^N$ and $\mathbf{A}_S(\Delta t) = \mathbf{A} \circ \mathbf{F}(\Delta t)$, has a complexity of $O(N^2)$ per input event if the base HiPPO matrix $\mathbf{A} \in \mathbb{R}^{N \times N}$ is dense. With a Normal-Plus-Low-Rank (NPLR) decomposition applied to $\mathbf{A}$, this complexity reduces to $O(Nr)$ per input event, where $r \ll N$ is the rank of the low-rank component, provided $\mathbf{F}(\Delta t)$ allows for efficient element-wise operations.*

This lemma highlights a cornerstone of FLAME's efficiency: the NPLR decomposition drastically reduces the computational cost of each state update from quadratic to near-linear ($O(Nr)$) in the state size $N$. This reduction is essential for real-time processing of potentially high-rate event data, as the core operations involving the adaptive state matrix $\mathbf{A}_S(\Delta t)$ remain computationally tractable.

## 4.2 Modeling Extended Temporal Contexts in Event Streams

**Theorem 1** (Preservation of Information over Extended Durations). *Let $\mathbf{x}(t) \in \mathbb{R}^N$ evolve according to the EA-HiPPO dynamics:*

$$\dot{\mathbf{x}}(t) = \mathbf{A}_S(\Delta t)\mathbf{x}(t) + \mathbf{B}\mathbf{E}_{flat}(t),$$

*where $\mathbf{A}_S(\Delta t) = \mathbf{A} \circ \mathbf{F}(\Delta t)$, $\mathbf{A}$ is a base HiPPO matrix with eigenvalues $\lambda_i(\mathbf{A})$ satisfying $Re(\lambda_i(\mathbf{A})) < 0$, $\mathbf{F}(\Delta t)$ is the adaptive decay matrix ($F_{ij}(\Delta t) = e^{-\alpha_{ij}\Delta t}$), inputs satisfy $\|\mathbf{E}_{flat}(t)\| \leq E_\infty$, and $\mathbf{x}(0) = \mathbf{x}_0$. If all eigenvalues $\lambda_k(\mathbf{A}_S(\Delta t))$ of the adaptive state matrix $\mathbf{A}_S(\Delta t)$ have negative real parts for relevant inter-event intervals $\Delta t$, then the state norm is bounded:*

$$\|\mathbf{x}(t)\| \leq e^{-\alpha_{eff}t}\|\mathbf{x}_0\| + \frac{\|\mathbf{B}\|E_\infty}{\alpha_{eff}}(1 - e^{-\alpha_{eff}t}),$$

*where $\alpha_{eff} = \min_k |Re(\lambda_k(\mathbf{A}_S(\Delta t)))| > 0$ is the memory retention rate, that adapts with $\Delta t$.*

Theorem 1 demonstrates FLAME's capacity to maintain information over long durations and understand temporally distant relationships within event streams. The EA-HiPPO dynamics allow the effective memory retention rate, $\alpha_{eff}$, to be small when necessary, preserving long-term context. Crucially, the adaptive nature of $\mathbf{A}_S(\Delta t)$, modulated by $\mathbf{F}(\Delta t)$ based on input event sparsity (inter-event interval $\Delta t$), allows this retention to be dynamically adjusted, aligning with the principles described in Sec. 3.

### 4.3 Numerical Stability of Event-Driven State Updates

**Lemma 2** (Stability of Taylor Expansion for State Update). *Approximating the matrix exponential* $e^{\mathbf{A}_S(\Delta t)\Delta t}$ *using an $n$-th order truncated Taylor series (Sec. 3), the single-step approximation error* $\mathbf{E}_n$ *is bounded by:*

$$\|\mathbf{E}_n\| \leq \frac{\|\mathbf{A}_S(\Delta t)\Delta t\|^{n+1}}{(n+1)!} e^{\|\mathbf{A}_S(\Delta t)\Delta t\|}.$$

This bound (Lemma 2) justifies using a low-order Taylor expansion (e.g., $n = 2$, as in Sec. 3) for the matrix exponential. This balances computational efficiency with numerical stability, ensuring reliable updates even with varying inter-event intervals $\Delta t$.

**Theorem 2** (Global System Stability). *The state trajectory $\mathbf{x}(t)$ of FLAME, governed by $\dot{\mathbf{x}}(t) = \mathbf{A}_S(\Delta t)\mathbf{x}(t) + \mathbf{B}\mathbf{E}_{flat}(t)$, remains bounded (i.e., $\|\mathbf{x}(t)\| \leq C$ for some constant $C > 0$, $\forall t \geq 0$) if:*

1. ***Bounded Inputs:*** *Input event streams ensure $\|\mathbf{E}_{flat}(t)\| \leq E_\infty$, $\forall t \geq 0$.*

2. ***Uniform Hurwitz Stability of*** $\mathbf{A}_S(\Delta t)$***:*** *For encountered inter-event intervals $\Delta t$, $\mathbf{A}_S(\Delta t)$ is uniformly Hurwitz (all eigenvalues $\lambda_k(\mathbf{A}_S(\Delta t))$ satisfy $Re(\lambda_k(\mathbf{A}_S(\Delta t))) \leq -\epsilon < 0$ for some fixed $\epsilon > 0$).*

3. ***Lyapunov Condition:*** *For each relevant $\mathbf{A}_S(\Delta t)$, $\exists$ a positive definite $\mathbf{P}(\Delta t)$ s.t. $\mathbf{A}_S(\Delta t)^T\mathbf{P}(\Delta t) + \mathbf{P}(\Delta t)\mathbf{A}_S(\Delta t) = -\mathbf{Q}(\Delta t)$ for some positive definite $\mathbf{Q}(\Delta t)$.*

Theorem 2 ensures the overall stability of FLAME. It guarantees that the internal state representation remains well-behaved and does not diverge, even when processing extended and sparse event streams, provided the adaptive state matrix $\mathbf{A}_S(\Delta t)$ maintains stability across the typical dynamics of inter-event intervals.

**Discussion: Balancing Efficiency and Memory for Event Data:** Our theoretical analysis highlights how FLAME is engineered for efficient event-based processing. The NPLR decomposition (Lemma 1) is fundamental to achieving low per-event update costs ($O(Nr)$), enabling the model to handle complex memory representations from event data without prohibitive computation. Concurrently, the EA-HiPPO mechanism's adaptive state matrix $\mathbf{A}_S(\Delta t)$ dynamically modulates the memory retention rate $\alpha_{eff}$ (Theorem 1) in response to input event sparsity, using the learnable decay matrix $\mathbf{F}(\Delta t)$. This creates an intrinsic balance between preserving information over extended temporal contexts and adapting to new, incoming events. Coupled with guarantees of numerical and global system stability (Lemma 2, Theorem 2), these theoretically-grounded design choices make FLAME particularly suitable for robust and resource-aware processing of high-dimensional event streams.

## 5 Experiments and Results

We conduct empirical evaluation of FLAME to demonstrate its effectiveness and efficiency in processing asynchronous, event-driven data from event cameras across various challenging benchmarks. Our experiments are designed to assess: (1) performance on demanding event-based vision datasets, including high-resolution and complex activity scenarios; (2) computational efficiency in terms of FLOPs, parameters, and inference latency, including profiling on diverse hardware backends; and (3) the specific contributions of key architectural components through targeted ablation studies.

### 5.1 Experimental Setup

**Datasets:** We evaluate FLAME on a diverse set of public benchmarks to assess its capabilities in event-based vision. For event-based vision, these include *DVS Gesture* [7] (gesture recognition), *HAR-DVS* [61] (human activity), *Celex-HAR* [2] (high-resolution human activity), *CIFAR10-DVS* [62] (neuromorphic image classification), and *N-Caltech101* [63] (neuromorphic object recognition). Additional evaluations cover event-based speech datasets, namely *Spiking Heidelberg Digits (SHD)* and *Spiking Speech Commands (SSC)* [64], and *Sequential CIFAR-10/100* [65] for sequential image classification, with further details provided in Appendix B.

Across all datasets, FLAME processes inputs event-by-event, dynamically updating its hidden state with each incoming event to preserve high temporal resolution. Comprehensive details on all dataset characteristics and specific task setups are available in Appendix B.1.

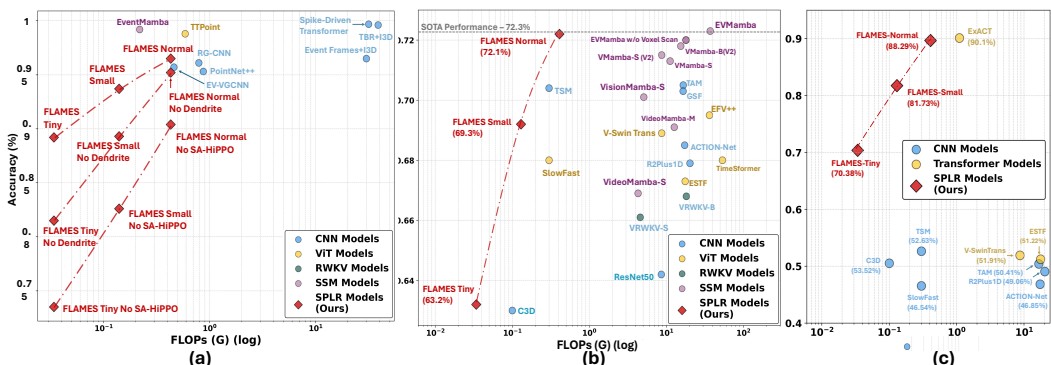

Figure 2: **Accuracy versus GFLOPs across various event-based vision datasets, comparing FLAME variants with other State-of-the-Art (SOTA) models.** (**a**) Performance on *DVSGesture128*, including ablation studies for FLAME demonstrating the impact of removing the Event Attention Layer (No Dendrite) or replacing EA-HiPPO with standard LIF neurons (No SA-HiPPO). (**b**) Performance on the high-resolution *CeleX-HAR* dataset, showcasing FLAME's efficiency at scale. (**c**) Performance on *HAR-DVS*, highlighting the competitive accuracy of FLAME variants. Note: FLAME variants consistently occupy favorable positions, indicating superior accuracy for their computational budget.

Table 2: Performance and Efficiency Comparison of FLAME-Normal against State-of-the-Art (SOTA) models across various event-based datasets. Dataset sizes are approximate. FLOPs are GigaFLOPs.

| Dataset | Resolution | Size (Samples; Classes) | SOTA Model (Acc. %) | SOTA FLOPs (G) | FLAME-Normal (Acc. %) | FLAME-Normal FLOPs (G) |
|---|---|---|---|---|---|---|
| CIFAR10-DVS [62] | 128 × 128 | ~10k Events/Img; 10 Cls | Spike-VGG11 (85.11%) | 8.9 | 80.5% | 0.43 |
| N-Caltech101 [63] | Orig: 302 × 245 (varies) | ~8.7k Samples; 101 Cls | EFV++ (89.7%) | - | 70.5% | 1.34 |
| DVS128 Gesture [7] | 128 × 128 | ~1.3k Samples; 11 Cls | EventMamba (99.2%) | 0.219 | 96.5% | 0.43 |
| HAR-DVS [61] | 346 × 260 | ~1.4k Samples; 6 Cls | EXACT (90.10%) | 1.3 | 88.29% | 0.41 |
| CeleX-HAR [2] | 1280 × 800 | ~125k Samples; 150 Cls | EVMamba (72.3%) | 37.2 | 72.2% | 0.41 |

**FLAME Model Variants:** We evaluate three primary variants of FLAME—Tiny, Small, and Normal—to demonstrate scalability and the trade-offs between performance and computational complexity. Their architectural details (e.g., number of distinct timing factors in the Event Attention Layer, filter counts, readout layer neurons) are provided in Suppl. Sec. C. All variants employ the EA-HiPPO mechanism.

**Evaluation Metrics and Implementation:** We report accuracy, GFLOPs (Giga Floating Point Operations), number of parameters, and inference latency (ms). FLAME models are implemented in PyTorch. Main training and inference for event datasets are conducted on NVIDIA A100 GPUs. Further details on hyperparameters and training procedures are in Appendix B.

## 5.2 Performance on Event-Based Vision Benchmarks

FLAME demonstrates strong performance across multiple event-based vision datasets, often achieving a superior accuracy-efficiency trade-off. This highlights its suitability for processing high-dimensional event camera data with low compute requirements.

**DVS Gesture:** As shown in Figure 2(a), FLAME-Normal obtains 96.5%, while FLAME-Small (93.7%) and FLAME-Tiny (89.2%) offer graceful performance degradation at lower compultation. These results compare favorably against models like EventMamba [40] and PointNet++ [66] but at much fewer FLOPS. Detailed comparisons in Suppl. Table 7.

**Contrast with EventMamba:** This comparison highlights the models' architectural focus. While EventMamba achieves slightly higher accuracy on the low-resolution DVS Gesture dataset (99.2% vs. FLAME's 96.5%) at comparable FLOPs, FLAME demonstrates its architectural advantage in processing high-resolution, sparse data on CeleX-HAR. Here, FLAME maintains comparable

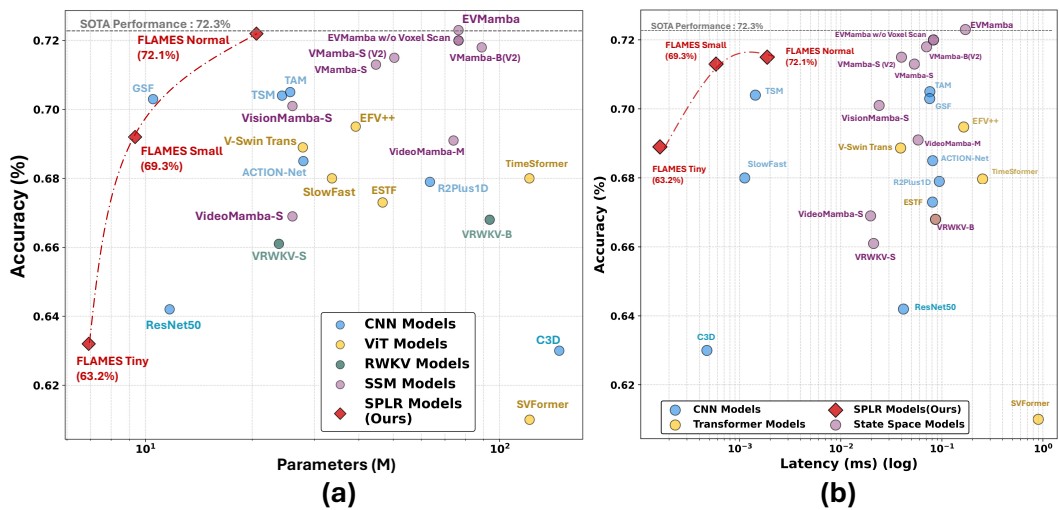

Figure 3: **Efficiency analysis on the CeleX-HAR dataset, measured on an NVIDIA A100 GPU.**
FLAME variants demonstrate a superior trade-off compared to SOTA models. (**a**) Accuracy versus
**Parameters (M)**: FLAME achieves competitive accuracy with significantly lower parameter counts
than many high-performance models. (**b**) Accuracy versus **Inference Latency (ms)** (log scale):
FLAME models exhibit substantially lower latency, confirming the efficiency of the asynchronous,
event-by-event design for real-time applications.

accuracy (72.2%) while operating at $\approx$ 90× lower computational cost (0.41 GFLOPs vs. 37.2
GFLOPs), showcasing superior efficiency and scalability. We detail this trade-off further in Appendix
B.4.

**HAR-DVS:** On this dataset for human activity recognition from event streams (Figure 2(c)), FLAME
shows strong performance, outperforming several state-of-the-art DNNs, with lower computational
cost.

**Celex-HAR (Scaling to HD Event Streams):** Figure 2(b) illustrates FLAME's scalability to
high-resolution (1280 × 800) event streams. FLAME-Normal surpasses the accuracy of models
like TSM [67] and VisionMamba-S [68] while operating with significantly lower GFLOPs. This
highlights EA-HiPPO's effectiveness in managing spatio-temporal complexity in high-dimensional
event data efficiently.

**Other Datasets:** FLAME also shows strong performance on other event datasets like CIFAR10-DVS,
NCaltech101, SHD and SSC event-based speech datasets as shown in Suppl. Table 6.

For completeness, we note that comparisons with voxel-based models like EventMamba are omitted
for datasets such as SHD and SSC because those models rely on frame-wise or voxel-based recon-
struction, which is not directly applicable to classification tasks where ground-truth labels are defined
at the sequence level.

### 5.3 Efficiency Analysis

**FLOPs and Parameters:** Figure 2 provides a comparative visualization of accuracy versus GFLOPs
across DVSGesture128, Celex-HAR, and HAR-DVS. FLAME variants consistently occupy favorable
positions, indicating superior accuracy for their computational budget compared to many established
models. For instance, on DVSGesture128, FLAME-Normal achieves high accuracy with significantly
fewer parameters and GFLOPs than several competing methods (further details in Appendix B).
The methodology for estimating FLOPs, particularly for our event-driven components, is detailed in
Appendix C.7.

**Inference Latency:** We measured inference latency on an NVIDIA A100 GPU (40GB VRAM).
As shown in Figure 3 for the challenging Celex-HAR dataset, FLAME models exhibit significantly
lower latency, crucial for real-time event processing. FLAME-Tiny achieves the lowest latency

at 0.162 ms. FLAME-Small (0.582 ms) and FLAME-Normal (1.867 ms) also maintain a strong latency-accuracy profile compared to models like TimeSformer (255.425 ms) and even optimized alternatives like VideoMamba-S (19.707 ms). Detailed latency figures are in Suppl. Table 8. To further assess practical deployability on diverse hardware, we profiled FLAME-Tiny across various platforms, including CPUs and other GPU tiers. These results, detailed in Appendix B.6 (Table 10), demonstrate consistent accuracy and highlight FLAME's adaptability.

**Ablation Studies:** To validate the contributions of FLAME's key architectural components to its overall performance and efficiency, we conducted ablation studies, primarily on the DVS Gesture 128 dataset (see Figure 2(a) dashed lines and Suppl. Table 7 for full details).
*Impact of Event Attention Layer:* Removing the multi-timescale processing mechanism within the Event Attention Layer resulted in a noticeable accuracy drop across all FLAME variants (e.g., FLAME-Normal accuracy decreased from 96.5% to 95.2%). This highlights the importance of this specialized initial processing stage for effectively capturing diverse temporal features present in raw event streams.
*Impact of EA-HiPPO:* Replacing the EA-HiPPO mechanism with standard LIF neurons ( keeping other architectural elements comparable for basic event processing) led to significant performance degradation (e.g., FLAME-Normal accuracy further reduced to 90.4%). This underscores the critical role of EA-HiPPO's structured, adaptive memory in retaining long-range temporal context from event data, a capability that simpler neuron models inherently lack for complex sequences.

These ablations confirm that both the Event Attention Layer and the EA-HiPPO mechanism are integral to FLAME's strong performance and its efficiency in processing event-based data.

# 6    Conclusion

In this paper, we introduced FLAME, a novel architecture engineered for the efficient processing of large-scale, asynchronous event-based data. By integrating an event-driven, neuromorphic-inspired front-end with a powerful state-space memory core, FLAME achieves compelling performance. Its Event-Aware HiPPO (EA-HiPPO) mechanism dynamically adapts memory based on inter-event intervals, enabling robust modeling of complex temporal patterns and the understanding of temporally distant relationships in sparse data streams. Our comprehensive evaluations demonstrate FLAME's significant advantages in accuracy, computational efficiency (low FLOPs/latency), and scalability across diverse benchmarks, including challenging event-based vision tasks and temporal reasoning, establishing it as a capable framework for processing real-world event data on conventional hardware.

**Current Limitations and Future Directions.** Despite its efficiency, the current FLAME framework has practical limitations. While it achieves high throughput on modern GPUs, it does not yet meet real-time constraints on CPU-only systems and is designed for static offline training, currently lacking mechanisms for on-device continual learning. Furthermore, as an event-driven model, its performance in high-noise environments may be sensitive to sensor noise, a known trade-off mitigated but not eliminated by the EAL filtering.

However, while FLAME incorporates event-driven principles, its core EA-HiPPO mechanism, as a sophisticated state-space model, is not a purely spiking neuromorphic algorithm in its current formulation. This means the full architecture is not directly translatable for end-to-end native execution on existing neuromorphic hardware platforms. A key direction for future work is therefore to adapt and extend the principles of FLAME, particularly its event-aware state dynamics, to enable efficient mapping onto such specialized hardware. Our goal is to leverage these advancements for ultra-low-power, large-scale, and efficient event-based vision at the edge, fully realizing the potential of neuromorphic computing for these demanding applications.

# Acknowledgements

This work is supported by the Defense Advanced Research Projects Agency (DARPA) under Grant Numbers N660012324003. The views and conclusions contained in this document are those of the authors and should not be interpreted as representing the official policies, either expressed or implied, of the Department of Defence, DARPA, or the U.S. Government.

## Broader Impacts

The primary societal impact of **FLAME** is derived from its architectural focus on **efficiency** and **real-time processing** for event-based vision. The core positive impact is enabling the deployment of sophisticated vision systems in **resource-constrained environments**. By achieving competitive accuracy on high-resolution data at substantially lower computational costs and ultra-low latency, this work significantly lowers the hardware and power barrier for applications in areas such as remote environmental monitoring, advanced robotics, and autonomous navigation at the edge.

As foundational architecture research that utilizes only established, publicly available benchmark datasets, this work introduces no unique negative societal risks beyond those generally associated with technological progress in machine perception. We acknowledge that enhanced real-time sensing capability could be potentially applied to advanced surveillance systems, but our focus remains on promoting the responsible use of this technology, specifically for resource-efficient and beneficial applications.

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

# A   Supplementary Section A: Detailed Proofs

## A.1   Computational Complexity of Event-Driven SSMs

**Lemma 1.** *Let the event-driven state-space model be governed by:*

$$\dot{x}(t) = Ax(t) + BS(t),$$

*where $x(t) \in \mathbb{R}^N$ is the internal state, $A \in \mathbb{R}^{N \times N}$ is the state transition matrix, and $S(t) \in \mathbb{R}^M$ is the input event train. The computational complexity of updating the internal state $x(t)$ at each event is $O(N^2)$.*

*Proof.* The event-driven state-space model is governed by:

$$\dot{x}(t) = Ax(t) + BS(t),$$

where $x(t) \in \mathbb{R}^N$ represents the internal state of the system, $A \in \mathbb{R}^{N \times N}$ is the state transition matrix, and $S(t) \in \mathbb{R}^M$ represents the input event train. When a event occurs at time $t_i$, the state update can be represented by the following integral equation for $t \in [t_i, t_{i+1})$:

$$x(t_i^+) = e^{A \Delta t_i} x(t_i^-) + \int_{t_i^-}^{t_i^+} e^{A(t_i^+ - \tau)} BS(\tau) \, d\tau,$$

where:

- $t_i^-$ and $t_i^+$ are the times just before and after the event at $t_i$,

- $\Delta t_i = t_i^+ - t_i^-$ is infinitesimal,

- $S(\tau)$ contains Dirac delta functions at event times and is zero elsewhere.

For simplicity, we focus on the update at the event time $t_i$ to approximate the state transition at each event.

The update of the internal state $x(t)$ requires computing the matrix exponential $e^{A \Delta t}$, where $\Delta t = t - t_i$ represents the time interval between successive events. Computing the exact matrix exponential for a general matrix $A \in \mathbb{R}^{N \times N}$ is computationally expensive, involving $O(N^3)$ operations using standard algorithms such as diagonalization or the Schur decomposition.

To reduce the computational cost, we approximate the matrix exponential using a truncated Taylor series expansion:

$$e^{A \Delta t_i} \approx I + A \Delta t_i + \frac{1}{2} A^2 \Delta t_i^2.$$

where $I$ is the identity matrix of size $N \times N$. This approximation is typically sufficient for small $\Delta t$, which is common between events.

In the Taylor series expansion approximation of $e^{A \Delta t}$, the dominant computational cost arises from multiplying the matrix $A \in \mathbb{R}^{N \times N}$ by itself and by the state vector $x(t) \in \mathbb{R}^N$.

The product $Ax(t)$, where $A \in \mathbb{R}^{N \times N}$ and $x(t) \in \mathbb{R}^N$, requires $N^2$ multiplications. Thus, the computational cost for this step is $O(N^2)$.

The term $A^2$ is computed by multiplying $A$ by itself. Since $A$ is an $N \times N$ matrix, computing $A^2$ explicitly would have a computational cost of $O(N^3)$. However, we avoid this by computing $A(Ax(t))$, which involves two sequential matrix-vector products, each costing $O(N^2)$. Therefore, the computational cost of computing $A^2 x(t)$ is $O(N^2)$.

The term $BS(t)$, where $B \in \mathbb{R}^{N \times M}$ and $S(t) \in \mathbb{R}^M$, involves $O(NM)$ operations. Assuming $M$ is proportional to $N$ or smaller, this computation contributes $O(N^2)$ to the overall complexity.

To update the internal state $x(t)$, we perform the following operations: First, we multiply $A$ by $x(t)$: $O(N^2)$; then multiply $A^2$ by $x(t)$: $O(N^2)$; followed by multiplying $B$ by $S(t)$: $O(NM)$ and finally add the resulting vectors.

Thus, the overall computational complexity for updating the internal state $x(t)$ at each event is $O(N^2)$.

In the general case, where $A$ is a dense matrix, the cost of updating the state is $O(N^2)$. If the matrix $A$ has a specific structure, such as being sparse or block-diagonal, the computational cost can be reduced. For example: - If $A$ is sparse with $k$ non-zero entries per row, the cost of multiplying $A$ by $x(t)$ becomes $O(kN)$, which can be significantly lower than $O(N^2)$ when $k \ll N$. - If $A$ is block-diagonal, the cost can be reduced to $O(N)$ per block, depending on the number and size of the blocks. However, for the general case where no such structure is assumed, the computational complexity remains $O(N^2)$. The computational complexity of updating the internal state $x(t)$ at each event, using the matrix exponential approximation with a Taylor series expansion, is dominated by the matrix-vector multiplication operations. Additionally, accounting for the $BS(t)$ term maintains the overall complexity at $O(N^2)$. Therefore, the overall computational complexity for updating the internal state at each event is $O(N^2)$.

$\square$

## A.2 Long-context Temporal Dependency Preservation Via Event-Based Hippo

**Theorem 1.** *Let $x(t) \in \mathbb{R}^N$ evolve according to*

$$\dot{x}(t) = Ax(t) + BS(t),$$

*where: - $A \in \mathbb{R}^{N \times N}$ is a HiPPO matrix with all eigenvalues satisfying $Re(\lambda_i) < 0$ for $i = 1, 2, \ldots, N$, - $B \in \mathbb{R}^{N \times M}$ is the input matrix, - $S(t) \in \mathbb{R}^M$ is the input event train, assumed to be bounded, i.e., there exists a constant $S_\infty > 0$ such that $\|S(t)\| \leq S_\infty$ for all $t \geq 0$, - $x_0 = x(0) \in \mathbb{R}^N$ is the initial state.*

*Then, the event-driven SSM preserves long-context temporal dependencies in the input event train $S(t)$, and the state $x(t)$ satisfies the bound:*

$$\|x(t)\| \leq e^{-\alpha t}\|x_0\| + \frac{\|B\|S_\infty}{\alpha}\left(1 - e^{-\alpha t}\right),$$

*where $\alpha = \min_i |Re(\lambda_i)| > 0$ is the memory retention factor determined by the eigenvalues of the HiPPO matrix $A$.*

*Proof.* To establish the theorem, we will analyze the evolution of the internal state $x(t)$ governed by the differential equation:

$$\dot{x}(t) = Ax(t) + BS(t),$$

with initial condition $x(0) = x_0$.

The differential equation is a non-homogeneous linear ordinary differential equation (ODE). Using the variation of parameters method, the solution can be expressed as:

$$x(t) = e^{At}x_0 + \int_0^t e^{A(t-\tau)}BS(\tau)\,d\tau,$$

where: - $e^{At}x_0$ is the solution to the homogeneous equation $\dot{x}(t) = Ax(t)$ with initial condition $x(0) = x_0$, - $\int_0^t e^{A(t-\tau)}BS(\tau)\,d\tau$ accounts for the particular solution due to the input $S(t)$.

Given that $A$ is a HiPPO matrix, all its eigenvalues satisfy $\text{Re}(\lambda_i) < 0$ for $i = 1, 2, \ldots, N$. This implies that $A$ is a Hurwitz matrix, ensuring that the system is asymptotically stable. Define the memory retention factor $\alpha$ as:

$$\alpha = \min_i |\text{Re}(\lambda_i)| > 0.$$

This factor dictates the rate at which the influence of the initial state $x_0$ decays over time.

Consider the homogeneous solution $e^{At}x_0$. Since all eigenvalues of $A$ have negative real parts, the matrix exponential $e^{At}$ satisfies:

$$\|e^{At}\| \leq e^{-\alpha t},$$

where $\|\cdot\|$ denotes an operator norm (e.g., the induced 2-norm). This inequality leverages the spectral bound of $A$ to provide an exponential decay rate.

Therefore, the contribution of the initial state is bounded by:

$$\|e^{At}x_0\| \leq \|e^{At}\| \cdot \|x_0\| \leq e^{-\alpha t}\|x_0\|.$$

Next, consider the particular solution:

$$\int_0^t e^{A(t-\tau)} BS(\tau)\,d\tau.$$

To bound its norm, apply the triangle inequality and properties of operator norms:

$$\left\| \int_0^t e^{A(t-\tau)} BS(\tau)\,d\tau \right\| \leq \int_0^t \| e^{A(t-\tau)} \| \cdot \| B \| \cdot \| S(\tau) \|\,d\tau.$$

Given that $\| S(\tau) \| \leq S_\infty$ and $\| e^{A(t-\tau)} \| \leq e^{-\alpha(t-\tau)}$, we have:

$$\left\| \int_0^t e^{A(t-\tau)} BS(\tau)\,d\tau \right\| \leq \| B \| S_\infty \int_0^t e^{-\alpha(t-\tau)}\,d\tau.$$

Evaluate the integral:

$$\int_0^t e^{-\alpha(t-\tau)}\,d\tau = \int_0^t e^{-\alpha s}\,ds = \frac{1 - e^{-\alpha t}}{\alpha}.$$

Thus, the bound becomes:

$$\left\| \int_0^t e^{A(t-\tau)} BS(\tau)\,d\tau \right\| \leq \frac{\| B \| S_\infty}{\alpha} \left( 1 - e^{-\alpha t} \right).$$

Combining the bounds for the homogeneous and particular solutions, we obtain:

$$\| x(t) \| \leq \| e^{At} x_0 \| + \left\| \int_0^t e^{A(t-\tau)} BS(\tau)\,d\tau \right\| \leq e^{-\alpha t} \| x_0 \| + \frac{\| B \| S_\infty}{\alpha} \left( 1 - e^{-\alpha t} \right).$$

This inequality demonstrates that the influence of the initial state $x_0$ decays exponentially at rate $\alpha$. Also, the accumulated influence of the input event train $S(t)$ is bounded and grows to a steady-state value determined by $\| B \|$, $S_\infty$, and $\alpha$.

The derived bound:

$$\| x(t) \| \leq e^{-\alpha t} \| x_0 \| + \frac{\| B \| S_\infty}{\alpha} \left( 1 - e^{-\alpha t} \right),$$

reveals that the term $e^{-\alpha t} \| x_0 \|$ signifies that the system "forgets" its initial state exponentially fast, ensuring that old information does not dominate the state indefinitely. Also, the integral term captures the accumulated influence of the input event train $S(t)$. Since $S(t)$ is bounded, the state $x(t)$ can retain and reflect information from the input over extended periods without being overwhelmed by the initial condition.

Therefore, the event-driven SSM governed by a HiPPO matrix $A$ effectively preserves long-context temporal dependencies in the input event train $S(t)$, while ensuring that the memory of the initial state $x_0$ decays at an exponential rate determined by $\alpha$.

$\square$

## A.3 Error Bound For Event-Driven Matrix Exponential Approximation

**Lemma 2.** *Let the matrix exponential be approximated using a Taylor expansion up to the $n$-th term:*

$$e^{A\Delta t} \approx I + A\Delta t + \frac{A^2 \Delta t^2}{2!} + \cdots + \frac{A^n \Delta t^n}{n!}.$$

*Assume that the matrix norm $\| \cdot \|$ is submultiplicative, i.e., $\| AB \| \leq \| A \| \| B \|$ for any matrices $A$ and $B$ of compatible dimensions. Then, the error $E_n$ of this approximation satisfies*

$$\| E_n \| \leq \frac{\| A\Delta t \|^{n+1}}{(n+1)!} e^{\| A\Delta t \|}.$$

*Proof.* The matrix exponential can be expressed as an infinite Taylor series:

$$e^{A\Delta t} = \sum_{k=0}^{\infty} \frac{(A\Delta t)^k}{k!}.$$

If we truncate this series after the $n$-th term, the remainder $E_n$ is given by:

$$E_n = e^{A\Delta t} - \sum_{k=0}^{n} \frac{(A\Delta t)^k}{k!} = \sum_{k=n+1}^{\infty} \frac{(A\Delta t)^k}{k!}.$$

To bound the norm of the error $E_n$, we apply the submultiplicative property of the matrix norm:

$$\|E_n\| = \left\| \sum_{k=n+1}^{\infty} \frac{(A\Delta t)^k}{k!} \right\| \leq \sum_{k=n+1}^{\infty} \frac{\|A\Delta t\|^k}{k!}.$$

Using the submultiplicative property of the matrix norm:

$$\|E_n\| \leq \sum_{k=n+1}^{\infty} \frac{\|A\Delta t\|^k}{k!}.$$

Let $x = \|A\Delta t\| \geq 0$. Then:

$$\|E_n\| \leq \sum_{k=n+1}^{\infty} \frac{x^k}{k!}.$$

Since

$$\sum_{k=n+1}^{\infty} \frac{x^k}{k!} = e^x - \sum_{k=0}^{n} \frac{x^k}{k!} = R_n(x),$$

where $R_n(x)$ is the remainder of the Taylor series expansion of $e^x$.

According to Taylor's Remainder Theorem (Lagrange's form), there exists $\xi \in [0, x]$ such that:

$$R_n(x) = \frac{x^{n+1}}{(n+1)!} e^{\xi}.$$

Since $\xi \leq x$ and $e^{\xi} \leq e^x$ for $x \geq 0$, we have:

$$R_n(x) \leq \frac{x^{n+1}}{(n+1)!} e^x.$$

Therefore:

$$\|E_n\| \leq \frac{x^{n+1}}{(n+1)!} e^x = \frac{\|A\Delta t\|^{n+1}}{(n+1)!} e^{\|A\Delta t\|}.$$

Thus, the error $E_n$ satisfies:

$$\|E_n\| \leq \frac{\|A\Delta t\|^{n+1}}{(n+1)!} e^{\|A\Delta t\|}.$$

$\square$

—

## A.4 Boundedness Of State Trajectories With Event Inputs

**Theorem 2.** *Boundedness of State Trajectory in Event-Driven State-Space Models*

*For a given initial condition $x_0$, the state trajectory $x(t)$ of the FLAME model driven by the event input $S(t)$ is bounded, i.e., $\|x(t)\| \leq C$, for some constant $C > 0$, provided that:*

1. *The input events $S(t)$ are of finite magnitude, i.e., $\|S(t)\| \leq S_\infty$ for all $t \geq 0$.*

2. *The decay matrix $A_S$ is Hurwitz, meaning all its eigenvalues have negative real parts.*

3. *There exists a positive definite matrix $P$ satisfying the Lyapunov equation $A_S^T P + P A_S = -Q$, for some positive definite matrix $Q$.*

*Proof.* Consider the FLAME governed by:

$$\dot{x}(t) = A_S x(t) + B S(t),$$

where $A_S$ is a Hurwitz matrix, $B$ is the input matrix, and $S(t)$ is a bounded input event train with $\|S(t)\| \leq S_\infty$ for all $t \geq 0$.

We define a Lyapunov function $V(x) = x^T P x$, where $P$ is a positive definite matrix satisfying the Lyapunov equation:

$$A_S^T P + P A_S = -Q,$$

with $Q$ being a positive definite matrix. Such a $P$ exists because $A_S$ is Hurwitz. The derivative of $V(x)$ along the system trajectories is computed:

$$\dot{V}(x) = \frac{d}{dt}(x^T P x) = x^T \dot{P} x + x^T P \dot{x} + \dot{x}^T P x.$$

Since $P$ is constant ($\dot{P} = 0$), and $\dot{x} = A_S x + B S(t)$, this simplifies to:

$$\dot{V}(x) = x^T P(A_S x + B S(t)) + (A_S x + B S(t))^T P x.$$

Recognizing that $P$ is symmetric ($P^T = P$), we can write:

$$\dot{V}(x) = x^T (A_S^T P + P A_S) x + 2 x^T P B S(t).$$

Substituting the Lyapunov equation $A_S^T P + P A_S = -Q$:

$$\dot{V}(x) = -x^T Q x + 2 x^T P B S(t).$$

The term $2 x^T P B S(t)$ is bounded using the Cauchy-Schwarz inequality as

$$2 x^T P B S(t) \leq 2\|x\| \cdot \|PB\| \cdot \|S(t)\| \leq 2\|PB\| S_\infty \|x\|.$$

Next, let us define $\gamma = 2\|PB\| S_\infty$ The derivative $\dot{V}(x)$ becomes:

$$\dot{V}(x) \leq -x^T Q x + \gamma \|x\|.$$

Since $Q$ is positive definite, $x^T Q x \geq \lambda_{\min}(Q) \|x\|^2$, where $\lambda_{\min}(Q)$ is the smallest eigenvalue of $Q$. Therefore:

$$\dot{V}(x) \leq -\lambda_{\min}(Q) \|x\|^2 + \gamma \|x\|.$$

Completing the square:

$$\dot{V}(x) \leq -\lambda_{\min}(Q)\left(\|x\|^2 - \frac{\gamma}{\lambda_{\min}(Q)}\|x\|\right) = -\lambda_{\min}(Q)\left(\|x\| - \frac{\gamma}{2\lambda_{\min}(Q)}\right)^2 + \frac{\gamma^2}{4\lambda_{\min}(Q)}.$$

This inequality indicates that $\dot{V}(x) < 0$ whenever $\|x\| > \frac{\gamma}{2\lambda_{\min}(Q)}$. Since $V(x) \geq 0$ and $\dot{V}(x)$ is negative outside a ball of radius $C = \frac{\gamma}{2\lambda_{\min}(Q)}$, the state $x(t)$ will ultimately remain within this bounded region. Therefore, $\|x(t)\| \leq C$ for all $t \geq 0$

$\square$

# B Supplementary Section B: Extended Experimental Results

## B.1 Datasets and Tasks

In this study, we evaluate the performance of the FLAME model across a diverse set of datasets, each presenting unique challenges in event-driven processing. The datasets include Sequential CIFAR-10, Sequential CIFAR-100 [65], DVS Gesture [7], HAR-DVS [61], Celex-HAR [2], Spiking Heidelberg Digits (SHD) [64], and Spiking Speech Commands (SSC). For all experiments, the FLAME model processes inputs on an event-by-event basis, leveraging its temporal dynamics to handle fine-grained temporal dependencies without accumulating events into frames. Below, we provide detailed descriptions of each dataset and the corresponding experimental setups.

**Sequential CIFAR-10 and CIFAR-100**: The CIFAR-10 and CIFAR-100 datasets [65] consist of $32 \times 32$ RGB images across 10 and 100 classes, respectively. To simulate a temporal sequence, each image is divided into 16 non-overlapping patches of size $8 \times 8$ pixels. These patches are presented to the model sequentially in a raster-scan order, from top-left to bottom-right. Each patch is treated as an independent event in the sequence. The task involves classifying the image based on the full sequence of patches, requiring the model to integrate information over the entire sequence. This setup evaluates the model's ability to process spatial information in a temporal context.

**DVS Gesture Dataset [7]:** This dataset comprises recordings of 11 hand gestures performed by 29 subjects under varying lighting conditions, captured by a Dynamic Vision Sensor (DVS). Event streams are processed at a resolution of $128 \times 128$ pixels. Sequences typically span approximately 1–6 seconds, and the maximum number of events processed per sequence is approximately 98k. FLAME processes each event individually as it occurs, without frame accumulation, to preserve high temporal resolution. The events in this dataset are highly sparse and asynchronous, with non-uniform distributions over the sequence duration.

**HAR-DVS Dataset [61]:** The HAR-DVS dataset contains neuromorphic event streams representing 6 human activities, such as walking and running. These sparse, asynchronous streams are recorded at a resolution of $346 \times 260$ pixels. Sequences generally last around 5 seconds, with the maximum number of events processed per sequence reaching approximately 450k. Our model processes each event-by-event, dynamically updating its internal state for each incoming event to precisely model activity sequences. The events are characterized by their spatial coordinates, timestamp, and polarity, exhibiting high sparsity and asynchronous, non-uniform temporal distribution.

**Celex-HAR Dataset [2]:** This dataset consists of high-resolution ($1280 \times 800$ pixels) event streams of human actions like sitting, standing, and walking, captured with a CeleX camera. Sequences mostly last around 2–3 seconds, with some extending beyond 5 seconds. The maximum number of events processed per sequence is up to approximately 3.1 million. FLAME processes each event-by-event, enabling it to capture the fine-grained temporal dynamics of these high-resolution human activities. Similar to other event-based datasets, the event data is highly sparse, asynchronous, and non-uniformly distributed over the sequence duration.

**Spiking Heidelberg Digits (SHD) and Spiking Speech Commands (SSC)**: The SHD and SSC datasets [64] contain neuromorphic event streams derived from speech datasets. SHD consists of spoken digit recordings converted to event trains using the CochleaAMS model, while SSC contains event-based representations of spoken command audio, representing keywords like "yes," "no," and "stop." Each event is characterized by its spatial location, timestamp, and polarity. The datasets evaluate the model's performance on tasks involving complex spatio-temporal patterns in speech data. The FLAME model processes each event as it occurs, dynamically updating its state, ensuring high temporal resolution and efficient processing for speech recognition tasks.

Across all datasets, the FLAME model processes inputs on an event-by-event basis. This approach allows it to maintain high temporal resolution and capture fine-grained spatio-temporal patterns, distinguishing it from frame-based methods. The event-by-event design also reduces computational overhead and ensures low latency, making the model well-suited for real-time applications.

## B.2 Ablation Studies:

To evaluate the contribution of individual components in the FLAME model, we performed extensive ablation studies on the sequential CIFAR-10 dataset. Specifically, we analyzed the impact of removing

Table 3: Comparison of FLAME models with state-of-the-art on the HARDVS dataset. Accuracy is measured in percentage, and computational cost is in GFLOPs.

| Model | GFLOPs | Accuracy (%) |
|---|---|---|
| C3D [69] | 0.1 | 50.52 |
| R2Plus1D [70] | 20.3 | 49.06 |
| TSM [67] | 0.3 | 52.63 |
| ACTION-Net [71] | 17.3 | 46.85 |
| TAM [72] | 16.6 | 50.41 |
| V-SwinTrans [73] | 8.7 | 51.91 |
| SlowFast [74] | 0.3 | 46.54 |
| ESTF [61] | 17.6 | 51.22 |
| ExACT [75] | 1.3 | 90.10 |
| **FLAME-Tiny** *[Ours]* | **0.034** | **65.42** |
| **FLAME-Small** *[Ours]* | **0.13** | **79.36** |
| **FLAME-Normal** *[Ours]* | **0.41** | **88.29** |

Table 4: Detailed Architecture of FLAME Models (Tiny, Small, and Normal)

| Layer Type | FLAME Tiny | FLAME Small | FLAME Normal |
|---|---|---|---|
| Input Representation | Asynchronous Event Events $(x, y, t, p)$ | | |
| Event Attention Layer | 16 attention branches $\tau_d = [\tau_1, \ldots, \tau_{16}]$ | 32 attention branches $\tau_d = [\tau_1, \ldots, \tau_{32}]$ | 64 attention branches $\tau_d = [\tau_1, \ldots, \tau_{64}]$ |
| Convolutional Block 1 | Conv2D (32 filters, 3x3) Batch Norm, Max Pool (2x2) | Conv2D (64 filters, 3x3) Batch Norm, Max Pool (2x2) | Conv2D (128 filters, 3x3) Batch Norm, Max Pool (2x2) |
| Convolutional Block 2 | Conv2D (32 filters, 3x3) Batch Norm, Max Pool (2x2) | Conv2D (64 filters, 3x3) Batch Norm, Max Pool (2x2) | Conv2D (128 filters, 3x3) Batch Norm, Max Pool (2x2) |
| Spatial Pooling Layer | Pool (2x2) | Pool (2x2) | Pool (2x2) |
| FLAME Convolution | State Update using Event-Aware HiPPO and NPLR decomposition for efficient event-driven convolution | | |
| Normalization Layer | Layer Norm | Layer Norm | Layer Norm |
| | Normalizes the state variables to stabilize training | | |
| Readout Layer | Fully Connected (256 neurons) Softmax for classification | Fully Connected (512 neurons) Softmax for classification | Fully Connected (1024 neurons) Softmax for classification |

or replacing key components such as the event attention layer, Event-Aware HiPPO (EA-HiPPO), NPLR decomposition, and FFT convolution. The results of these experiments, along with the corresponding model parameters and computational costs (in GFLOPs), are summarized in Table 7.

**Impact of Event Attention Layer (EAL)** Removing the event based attention mechanism leads to a reduction in both accuracy and model parameters. The accuracy drops across all channel configurations, with the largest channels (128) seeing a decrease from 90.25% to 85.83%. The smaller channel configurations (64 and 32) experience similar drops, highlighting the event based

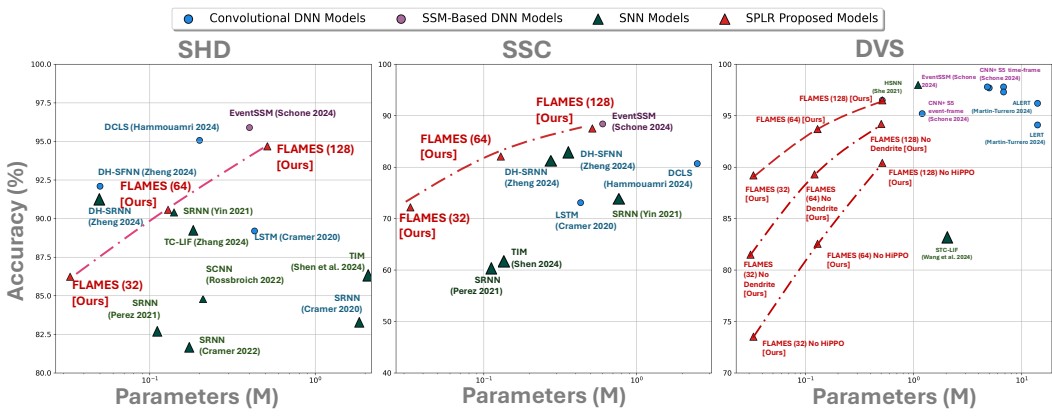

Figure 4: Comparison of our FLAME to the state-of-the-art on DVS128-Gesture[7], Spiking Heidelberg digits (SHD) and Spiking Speech Commands (SSC) [64] datasets

Table 5: Experimental results on CeleX-HAR dataset.

| No. | Algorithm | Publish | Arch. | FLOPs | Params | acc/top-1 | Code |
|-----|-----------|---------|-------|-------|--------|-----------|------|
| 01 | ResNet-50 [76] | CVPR-2016 | CNN | 8.6G | 11.7M | 0.642 | URL |
| 02 | ConvLSTM [77] | NIPS-2015 | CNN, LSTM | - | - | 0.539 | URL |
| 03 | C3D [69] | ICCV-2015 | CNN | 0.1G | 147.2M | 0.630 | URL |
| 04 | R2Plus1D [70] | CVPR-2018 | CNN | 20.3G | 63.5M | 0.679 | URL |
| 05 | TSM [67] | ICCV-2019 | CNN | 0.3G | 24.3M | 0.704 | URL |
| 06 | ACTION-Net [71] | CVPR-2021 | CNN | 17.3G | 27.9M | 0.685 | URL |
| 07 | TAM [72] | ICCV-2021 | CNN | 16.6G | 25.6M | 0.705 | URL |
| 08 | GSF [78] | TPAMI-2023 | CNN | 16.5G | 10.5M | 0.703 | URL |
| 09 | V-SwinTrans [73] | CVPR-2022 | ViT | 8.7G | 27.8M | 0.689 | URL |
| 10 | TimeSformer [79] | ICML-2021 | ViT | 53.6G | 121.2M | 0.680 | URL |
| 11 | SlowFast [74] | ICCV-2019 | ViT | 0.3G | 33.6M | 0.680 | URL |
| 12 | SVFormer [80] | CVPR-2023 | ViT | 196.0G | 121.3M | 0.610 | URL |
| 13 | EFV++ [81] | arXiv-2024 | ViT, GNN | 36.3G | 39.2M | 0.695 | URL |
| 14 | ESTF [61] | AAAI-2024 | ViT, CNN | 17.6G | 46.7M | 0.673 | URL |
| 15 | VRWKV-S [82] | arXiv-2024 | RWKV | 4.6G | 23.8M | 0.661 | URL |
| 16 | VRWKV-B [82] | arXiv-2024 | RWKV | 18.2G | 93.7M | 0.668 | URL |
| 17 | Vision Mamba-S [68] | ICML-2024 | SSM | 5.1G | 26.0M | 0.701 | URL |
| 18 | VMamba-S [83] | arXiv-2024 | SSM | 11.2G | 44.7M | 0.713 | URL |
| 19 | VMamba-S(V2) [83] | arXiv-2024 | SSM | 8.7G | 50.4M | 0.715 | URL |
| 20 | VMamba-B [83] | arXiv-2024 | SSM | 18.0G | 76.5M | 0.720 | URL |
| 21 | VMamba-B(V2) [83] | arXiv-2024 | SSM | 15.4G | 88.9M | 0.718 | URL |
| 22 | VideoMamba-S [84] | ECCV-2024 | SSM | 4.3G | 26.0M | 0.669 | URL |
| 23 | VideoMamba-M [84] | ECCV-2024 | SSM | 12.7G | 74.0M | 0.691 | URL |
| 24 | EVMamba | arXiv-2024 | SSM | 37.2G | 76.5M | **0.723** | URL |
| 25 | EVMamba *w/o* Voxel Scan | arXiv-2024 | SSM | 18.0G | 76.5M | 0.720 | URL |
| 26 | **FLAME-Tiny (Ours)** | - | SSM | 0.034G | 7.91M | 0.632 | - |
| 27 | **FLAME-Small (Ours)** | - | SSM | 0.13G | 13.35M | 0.692 | - |
| 28 | **FLAME-Normal (Ours)** | - | SSM | 0.41G | 25.57M | 0.722 | - |

attention's role in improving the spatio-temporal feature representation. Interestingly, removing this mechanism slightly reduces the model's GFLOPs since the computations associated with the event attention layer are avoided.

**Impact of Event-Aware HiPPO** Replacing EA-HiPPO with a simple LIF-based mechanism leads to a moderate drop in accuracy (e.g., from 90.25% to 87.62% for 128 channels). However, this modification does not alter the computational cost (GFLOPs), as EA-HiPPO primarily affects the temporal memory adaptation rather than the core matrix or convolution operations. These results emphasize EA-HiPPO's critical role in retaining and managing temporal dynamics effectively.

**Impact of NPLR Decomposition** The NPLR decomposition significantly reduces the computational complexity of state-space updates. For the FLAME-Normal (128 channels) model, removing NPLR also resulted in a notable increase in GPU memory usage by approximately 38% (from 5.00 MB to 6.91 MB), confirming NPLR's essential role in minimizing both computational cost and memory footprint for real-time deployment. Despite this computational overhead, the accuracy remains relatively stable, highlighting that NPLR's primary advantage is computational efficiency rather than feature extraction performance.

**Impact of FFT Convolution** FFT convolution is integral to efficiently handling long-context temporal dependencies. Replacing FFT convolution with standard time-domain convolution increases the GFLOPs substantially (e.g., from 0.43 GFLOPs to 1.2 GFLOPs for 128 channels). Furthermore, the accuracy sees a more pronounced decline (e.g., from 90.25% to 86.47%), particularly in tasks requiring high temporal resolution. These results underscore FFT convolution's dual role in reducing computational cost and maintaining temporal modeling performance.

**Summary of Findings** The ablation studies validate the critical importance of each component in the FLAME model:

- The event attention layer enhances the spatio-temporal feature representation, significantly improving accuracy.

Table 6: Comparison of classification accuracy and parameters of different models across SHD and SSC datasets.

| Model | SHD | | SSC | |
|---|---|---|---|---|
| | #Parameters | Accuracy (%) | #Parameters | Accuracy (%) |
| SFNN [64] | 0.09 M | 48.1 | 0.09 M | 32.5 |
| SRNN [64] | 1.79 M | 83.2 | - | - |
| SRNN [85] | 0.17 M | 81.6 | - | - |
| SRNN [86] | 0.11 M | 82.7 | 0.11 M | 60.1 |
| SCNN [87] | 0.21 M | 84.8 | - | - |
| SRNN [88] | 0.14 M | 90.4 | 0.77 M | 74.2 |
| HRSNN [89] | - | 80.01 | - | 59.28 |
| LSTM [64] | 0.43 M | 89.2 | 0.43 M | 73.1 |
| DH-SRNN [33] | 0.05 M | 91.34 | 0.27 M | 81.03 |
| DH-SFNN [33] | 0.05 M | 92.1 | 0.35 M | 82.46 |
| ASGL [90] | - | 78.90 | - | 78.90 |
| DCLS [91] | 0.2 M | 95.07 | 2.5 M | 80.69 |
| TIM [92] | 2.59 M | 86.3 | 0.111 M | 61.09 |
| TC-LIF [93] | 0.142 M | 88.91 | - | - |
| FLAME-Normal (128) [Ours] | 0.513 M | 94.68 | 0.513 M | 87.52 |
| FLAME-Small (64) [Ours] | 0.129 M | 90.57 | 0.129 M | 82.08 |
| FLAME-Tiny (32) [Ours] | 0.033 M | 86.24 | 0.033 M | 72.19 |

- EA-HiPPO dynamically adjusts temporal memory retention, contributing to performance robustness without additional computational overhead.

- NPLR decomposition ensures scalability by reducing the computational cost of state-space updates, making the model efficient for large-scale tasks.

- FFT convolution is indispensable for capturing long-context dependencies efficiently while keeping computational complexity low.

## B.3 DVS Gesture Recognition

To further investigate the combined effectiveness of Event-based Attention mechanisms and *FLAME* convolutions in event-based processing, we evaluate our model on the *DVS Gesture* dataset. This dataset consists of event streams recorded from a Dynamic Vision Sensor (DVS) at a resolution of $128 \times 128$, providing a challenging benchmark for evaluating temporal dynamics in gesture recognition tasks involving varying speeds and motions.

Our goal is to assess how the integration of event-based attention mechanisms with *FLAME* convolution layers enhances the model's ability to capture multi-scale temporal dependencies. Specifically, we examine how event-based attention can serve as a temporal attention mechanism that helps *FLAME* effectively focus on the most relevant events, while *FLAME* convolutions manage the overall temporal and spatial evolution of features.

The experiment involves training variants of our model—one incorporating both event-based attention mechanisms and *FLAME* convolutions, and the other using only *FLAME*—to determine the contribution of event-based attention . Table 7 summarizes the test accuracy of our models compared to other state-of-the-art approaches. The results are measured in terms of classification accuracy, along with the number of parameters, to highlight model efficiency.

As shown in Table 7, the *FLAME* model with 128 channels, incorporating event-based attention , achieves 96.5% accuracy while maintaining a significantly lower parameter count compared to many other state-of-the-art models. This shows that our approach effectively utilizes sparse event-driven inputs to achieve high accuracy with reduced computational complexity. The use of event-based attention mechanisms allows the model to dynamically adjust its focus on different temporal scales, thus improving gesture recognition even in scenarios with rapid motion changes.

The variant without event-based attention , while still competitive, lags behind in adapting to the multi-scale nature of the event data, especially for gestures with complex temporal characteristics. This indicates that the event-based attention mechanism plays a crucial role in adaptively filtering

Table 7: Comparison of classification accuracy, parameters, and FLOPs of different models across the DVS128-Gesture dataset.

| Model | #Parameters (M) | GFLOPs | Accuracy (%) |
|---|---|---|---|
| Yousefzadeh et al. [94] | 1.2 | - | 95.2 |
| Xiao et al. [95] | - | - | 96.9 |
| RTRL [96] | 4.8 | - | 97.8 |
| She et al. [97] | 1.1 | - | 98.0 |
| Liu et al. [98] | - | - | 98.8 |
| Chakraborty et al. [27] | - | - | 96.5 |
| Martin-Turrero et al. [15] | 14 | - | 96.2 |
| Martin-Turrero et al. [15] | 14 | - | 94.1 |
| CNN + S5 (time-frames) [99] | 6.8 | - | 97.8 |
| Event-SSM [99] | 5 | - | 97.7 |
| CNN + S5 (event-frames) [99] | 6.8 | - | 97.3 |
| TBR+I3D [100] | 12.25 | 38.82 | 99.6 |
| Event Frames + I3D [12] | 12.37 | 30.11 | 96.5 |
| EV-VGCNN [13] | 0.82 | 0.46 | 95.7 |
| RG-CNN [101] | 19.46 | 0.79 | 96.1 |
| PointNet++ [102] | 1.48 | 0.872 | 95.3 |
| PLIF [103] | 1.7 | - | 97.6 |
| GET [104] | 4.5 | - | 97.9 |
| Swin-T v2 [105] | 7.1 | - | 93.2 |
| TTPOINT [40] | 0.334 | 0.587 | 98.8 |
| EventMamba [40] | 0.29 | 0.219 | 99.2 |
| STC-LIF [106] | 3.922 | - | 83.0 |
| Spiking Transformer [14] | 36.01 | 33.32 | 99.3 |
| FLAME-Normal (128) **[Ours]** | 0.513 | 0.43 | 96.5 |
| FLAME-Small (64) **[Ours]** | 0.129 | 0.14 | 93.7 |
| FLAME-Tiny (32) **[Ours]** | 0.033 | 0.07 | 89.2 |
| FLAME-Normal (128 Channels) No Attention Branch **[Ours - Ablation]** | 0.501 | 0.43 | 95.2 |
| FLAME-Small (64 Channels) No Attention Branch **[Ours - Ablation]** | 0.121 | 0.14 | 89.3 |
| FLAME-Tiny (32 Channels) No Attention Branch **[Ours - Ablation]** | 0.031 | 0.07 | 81.5 |
| FLAME-Normal (128 Channels) No HiPPO **[Ours - Ablation]** | 0.501 | 0.43 | 90.4 |
| FLAME-Small (64 Channels) No HiPPO **[Ours - Ablation]** | 0.121 | 0.14 | 82.6 |
| FLAME-Tiny (32 Channels) No HiPPO **[Ours - Ablation]** | 0.031 | 0.07 | 73.5 |

relevant temporal features, which is essential for handling the asynchronous, irregular inputs typical of event cameras.

In addition, visualizations of the learned event-based attention reveal how the model attends to different time segments, effectively filtering the incoming event streams to prioritize the most relevant events. This adaptive filtering complements the *FLAME* convolutional operations, leading to more robust and efficient temporal feature extraction.

Overall, the results validate the utility of combining event-based attention mechanisms with *FLAME* convolutions for event-driven tasks, making the model well-suited for gesture recognition from DVS inputs. The joint use of these components allows for efficient temporal modeling, maintaining a favorable trade-off between accuracy and parameter efficiency.

## B.4 Scaling to HD Event Streams

The scalability of the proposed *FLAME* model is evaluated on the *Celex HAR* dataset, a human activity recognition dataset recorded at a high resolution of $1280 \times 800$. This dataset serves as a challenging benchmark for assessing the model's ability to maintain high accuracy and computational efficiency when processing large-scale spatial and temporal data.

In this experiment, *FLAME* is used for action recognition on HD event streams, and its performance is compared to that of baseline Spiking Neural Networks (SNNs) and State-Space Models (SSMs). As shown in Figure 2, the results demonstrate that *FLAME* maintains high accuracy even at increased resolutions, whereas the baseline models experience significant performance degradation due to heightened computational demands. The integration of the *FLAME convolution layer* proves effective in managing the complex spatial and temporal components of HD event data, providing robust real-time processing capabilities with minimal computational overhead.

Figure 2 illustrates the trade-off between accuracy and computational cost, measured in terms of FLOPs, for our *FLAME* models compared to state-of-the-art methods on the *Celex-HAR* dataset. The *FLAME* variants—*FLAME Tiny*, *FLAME Small*, and *FLAME Normal*—demonstrate superior effi-

Table 8: Latency Comparison on Celex-HAR (in milliseconds)

| Algorithm | Latency (ms) | Algorithm | Latency (ms) | Algorithm | Latency (ms) |
|---|---|---|---|---|---|
| **FLAME-Tiny** | **0.162** | TAM | 76.012 | VideoMamba-S | 19.707 |
| **FLAME-Small** | **0.582** | GSF | 75.558 | VideoMamba-M | 58.164 |
| **FLAME-Normal** | **1.867** | V-SwinTrans | 39.837 | EVMamba | 170.34 |
| ResNet-50 | 41.575 | TimeSformer | 255.425 | EVMamba w/o Voxel Scan | 82.423 |
| C3D | 0.473 | SlowFast | 1.118 | VRWKV-S | 21.091 |
| R2Plus1D | 94.264 | EFV++ | 166.23 | VRWKV-B | 86.346 |
| TSM | 1.4266 | ESTF | 80.61 | Vision Mamba-S | 23.88 |
| ACTION-Net | 81.035 | SVFormer | 897.455 | VMamba-S | 53.302 |
| TAM | 76.012 | | | VMamba-S(V2) | 39.848 |
| GSF | 75.558 | | | VMamba-B | 82.421 |
| V-SwinTrans | 39.837 | | | VMamba-B(V2) | 70.514 |

ciency by achieving competitive or better accuracy while utilizing significantly fewer computational resources.

Key observations from Figure 2 are as follows:

- **Efficiency at Different Scales**: *FLAME Tiny* achieves approximately 63.8% accuracy with a fraction of the computational cost compared to larger models such as *SlowFast* and *C3D*. As the model scales to *FLAME Small* and *FLAME Normal*, accuracy improves to 69.3% and 72.1%, respectively, while maintaining a favorable computational cost profile.

- **Performance with Reduced Complexity**: *FLAME Normal* matches or exceeds the accuracy of models like *TSM* and *VisionMamba-S* but at a substantially lower computational cost. This efficiency is attributed to the integration of event-driven processing and effective state-space dynamics.

The improved efficiency of *FLAME* can be credited to the event-based processing capabilities of the **FLAME architecture** and the **FLAME convolution layer**, which optimally manage state-space evolution without relying on dense operations. These features allow the model to capture complex temporal dependencies while minimizing computational requirements, making *FLAME* particularly effective for high-resolution event-based datasets like *Celex-HAR*.

**HAR-DVS Results:** The HAR-DVS dataset results underscore the advantages of our FLAME models, achieving accuracies of **70.38%**, **81.73%**, and **88.29%** for FLAME-Tiny, FLAME-Small, and FLAME-Normal, respectively, while maintaining substantially lower computational costs compared to other state-of-the-art models. Unlike traditional deep neural networks such as C3D and R2Plus1D, which struggle to model the complex temporal relationships inherent in event streams, FLAME leverages a novel *event-by-event processing approach*, preserving fine-grained temporal dynamics essential for accurate action recognition.

Moreover, FLAME employs a unique *event-based attention mechanism* that enhances its ability to capture long-context spatio-temporal dependencies efficiently. The prolonged and complex actions in HAR-DVS demand robust temporal attention mechanisms, as highlighted in prior studies. FLAME's dendritic-inspired design meets these requirements while offering a computationally efficient solution, making it particularly suitable for real-time, low-latency applications in dynamic event-driven environments.

It is important to note that HAR-DVS provides frame-based data, as raw event data was unavailable for download. Since FLAME is designed for event-by-event processing, we treated all events arriving at the same timestamp as a single batch for processing, adhering to the event-driven principles of the model.

## B.5 Performance on N-Caltech101

To further assess the generalization capabilities of FLAME on object recognition tasks using event-based data, we evaluated its performance on the N-Caltech101 dataset. The results for FLAME-Tiny, FLAME-Small, and FLAME-Normal are presented in Table 9.

Table 9: Performance of FLAME Variants on N-Caltech101.

| Model Variant | Accuracy (%) | Params (M) | FLOPs (G) |
|---|---|---|---|
| FLAME-Tiny | 65.9 | 0.033 | 0.71 |
| FLAME-Small | 68.3 | 0.129 | 0.86 |
| FLAME-Normal | 70.5 | 0.513 | 1.34 |

The results demonstrate that FLAME scales effectively with model capacity on this challenging classification benchmark, achieving competitive accuracy while maintaining a low computational footprint. This reinforces its suitability for real-world event-based learning scenarios involving diverse object categories.

## B.6 Hardware Deployment and Efficiency Profile

To assess the practical deployability and efficiency of FLAME beyond standard GPU benchmarks, we profiled the FLAME-Tiny variant across a diverse range of hardware platforms. This evaluation aims to demonstrate FLAME' adaptability to both high-performance computing environments and more resource-constrained settings, which is crucial for real-world applications of event-based systems. The FLAME-Tiny model was selected for this profiling due to its design emphasis on minimal computational footprint, making it a suitable candidate for edge deployment scenarios.

The performance of FLAME-Tiny, in terms of inference accuracy and latency, was measured on several common hardware backends, including a standard CPU and various NVIDIA GPUs. The results, obtained without any model retraining specifically for these hardware targets, are summarized in Table 10.

Table 10: FLAME-Tiny Inference Performance on Diverse Hardware Platforms. Accuracy was evaluated on the Celex-HAR dataset. Latency is reported in milliseconds (ms).

| Hardware | Accuracy (%) | Latency (ms) |
|---|---|---|
| Intel Xeon CPU (2 vCPUs @2.2GHz) | 63.2 | 42.1 |
| NVIDIA T4 GPU | 63.2 | 1.57 |
| NVIDIA V100 GPU (32GB) | 63.2 | 0.43 |
| NVIDIA A100 GPU (40GB) | 63.2 | 0.162 |

The results in Table 10 indicate that FLAME-Tiny maintains its accuracy (63.2% on Celex-HAR) consistently across all tested hardware. As expected, the inference latency varies significantly, with the NVIDIA A100 GPU providing the fastest processing time at 0.162 ms. Even on a CPU, FLAME-Tiny achieves a reasonable latency of 42.1 ms, demonstrating its potential for deployment in environments where specialized GPU hardware may not be available. The performance on the NVIDIA T4, a GPU often used for inference in datacenter and edge scenarios, also shows a practical latency of 1.57 ms.

These findings support the claim that FLAME, particularly its lightweight variants like FLAME-Tiny, can be efficiently deployed across a spectrum of computational platforms, from high-end GPUs to more general-purpose CPUs, without compromising accuracy. This versatility is a key advantage for event-based processing, where applications can range from power-constrained mobile devices to high-throughput server-side systems.

# C    Supplementary Section C: Methods and Architectural Details

---

**Algorithm 1** FLAME Model Training

---

**Require:** Training dataset $\mathcal{D} = \{(\mathbf{X}_i, \mathbf{y}_i)\}_{i=1}^{N}$, learning rate $\eta$, total epochs $E$, threshold potential $V_{\text{th}}$, decay factors $\alpha_d$, $\beta$

1: Initialize weights $\mathbf{W}$, event-based attention timing factors $\tau_d$, FLAME matrices $\mathbf{A}, \mathbf{B}, \mathbf{C}$, low-rank matrices $\mathbf{P}, \mathbf{Q}$, and kernel $\mathbf{K}(\omega)$

2: Initialize coupling strengths $\mathbf{g}_d$ for each event attention branch $d$

3: **for** epoch = 1 to $E$ **do**

4:     **for** each $(\mathbf{X}, \mathbf{y}) \in \mathcal{D}$ **do**

    **Input Representation:** Prepare input events for processing

5:         Parse input event sequence $\mathbf{X} = \{(x_i, y_i, t_i, p_i)\}$, where $(x_i, y_i)$ are spatial coords, $t_i$ is time, $p_i$ is polarity.

    **Event Attention Layer:** Update attention branch currents and aggregate at soma

6:         **for** each $t_i$ in event sequence **do**

7:             **for** each attention branch $d$ **do**

8:                 Update attention branch current: $\mathbf{i}_d(t_i + 1) = \alpha_d \cdot \mathbf{i}_d(t_i) + \sum_{j \in \mathcal{N}_d} \mathbf{w}_j \cdot p_j$

9:             **end for**

10:             Aggregate currents at soma: $V(t_i + 1) = \beta \cdot V(t_i) + \sum_d \mathbf{g}_d \cdot \mathbf{i}_d(t_i)$

11:             **if** $V(t_i + 1) > V_{\text{th}}$ **then**

12:                 Generate event and reset potential: $V(t_i + 1) \leftarrow 0$

13:             **end if**

14:         **end for**

    **Spatial Pooling Layer:** Reduce spatial dimensionality while preserving temporal resolution

15:         Apply max pooling: $I_{\text{pooled}}(x', y', t) = \max_{(x,y) \in P(x',y')} I(x, y, t)$

    **FLAME Conv. Layer:** Apply EA-HiPPO, NPLR, & FFT for event dynamics

16:         Initialize state vector $\mathbf{x}(0)$

17:         **for** each event time $t_k$ in $I_{\text{pooled}}$ **do**

18:             Compute $\Delta t_k = t_{k+1} - t_k$, decay $\mathbf{F}_{ij}(\Delta t_k) = e^{-\alpha_{ij} \cdot \Delta t_k}$

19:             Compute event-aware HiPPO: $\mathbf{A_S} = \mathbf{A} \circ \mathbf{F}(\Delta t_k)$

20:             Decompose: $\mathbf{A_S} = \mathbf{V}\mathbf{\Lambda}\mathbf{V}^* - \mathbf{P}\mathbf{Q}^*$

21:             $e^{\mathbf{A_S}\Delta t_k} \approx \mathbf{I} + \mathbf{A_S}\Delta t_k + \dfrac{\mathbf{A_S}^2(\Delta t_k)^2}{2}$

22:             Update: $\mathbf{x}(t_{k+1}) = e^{\mathbf{A_S}\Delta t_k} \cdot \mathbf{x}(t_k) + \mathbf{A_S}^{-1}(e^{\mathbf{A_S}\Delta t_k} - \mathbf{I}) \cdot \mathbf{B} \cdot \mathbf{S}(t_k)$

23:             FFT-based convolution: $\mathbf{x}(t_{k+1}) = \text{IFFT}(\text{FFT}(\mathbf{K}(\omega)) \odot \text{FFT}(\mathbf{x}(t_{k+1})))$

24:         **end for**

25:         Compute continuous output: $\mathbf{y}(t) = \mathbf{C} \cdot \mathbf{x}(t)$

26:         **Thresholding:** Convert $\mathbf{y}(t)$ to events by applying $y_{\text{event}}(t) = \mathbb{I}(\mathbf{y}(t) > V_{\text{th}})$

    **Normalization:** Reduce variability in activations

27:         Apply layer normalization: $\hat{\mathbf{x}}_l = \dfrac{\mathbf{x}_l - \mu_l}{\sqrt{\sigma_l^2 + \epsilon}} \cdot \gamma + \beta$

    **Readout Layer:** Compute final output and update model parameters

28:         Compute pooled state: $\mathbf{x}_{\text{pooled},k} = \frac{1}{p} \sum_{i=kp}^{(k+1)p-1} \hat{\mathbf{x}}_i$

29:         Final output: $\mathbf{y}_{\text{pred}} = \mathbf{W} \cdot \mathbf{x}_{\text{pooled}} + \mathbf{b}$

30:         Compute loss $\mathcal{L}(\mathbf{y}_{\text{pred}}, \mathbf{y})$, update $\mathbf{W} \leftarrow \mathbf{W} - \eta \cdot \frac{\partial \mathcal{L}}{\partial \mathbf{W}}$

31:     **end for**

32: **end for**

---

**Background and Preliminaries**

**State-Space Models:** A state-space model (SSM) is a mathematical framework for modeling systems that evolve over time. The dynamics of such systems are described by a set of first-order differential equations, often expressed in continuous time as:

$$\dot{x}(t) = Ax(t) + Bu(t), \quad y(t) = Cx(t) + Du(t)$$

where:

- $x(t) \in \mathbb{R}^N$ is the hidden state vector, representing the internal state of the system at time $t$,

- $u(t) \in \mathbb{R}^M$ is the input signal, such as sensory data or external stimuli,
- $y(t) \in \mathbb{R}^P$ is the output signal or observable state,
- $A \in \mathbb{R}^{N \times N}$, $B \in \mathbb{R}^{N \times M}$, $C \in \mathbb{R}^{P \times N}$, and $D \in \mathbb{R}^{P \times M}$ are learned system matrices.

State-space models are often used in signal processing and control systems to model systems with temporal dependencies. In many practical scenarios, however, the continuous-time formulation is discretized:

$$x_{k+1} = A_d x_k + B_d u_k, \quad y_k = C_d x_k + D_d u_k$$

where $A_d$, $B_d$, $C_d$, and $D_d$ are the corresponding discrete-time matrices, and $k$ indexes the discrete time steps.

**Highly Optimized Polynomial Projection (HiPPO):** The *HiPPO* framework provides a method for approximating the continuous history of an input signal by projecting it onto a set of polynomial basis functions. The HiPPO matrix $A$ is designed to optimally compress the history of the input into a state vector $x(t)$, allowing the model to retain relevant temporal dependencies over long time scales. For example, the HiPPO-Legendre (HiPPO-LegS) matrix $A$ is defined as:

$$A_{nk} = \begin{cases} -\sqrt{(2n+1)(2k+1)} & \text{if } n > k \\ n+1 & \text{if } n = k \\ 0 & \text{if } n < k \end{cases}$$

This matrix governs the dynamics of how the internal state evolves to represent the history of the input in a compressed manner.

**Mathematical Modeling and Event Generation Mechanism**

Events in the FLAME model are generated through the dynamics of LIF neurons. The event generation process is described in detail below:

- **Current Integration in Event-based attention branch** : Each DH-LIF neuron integrates incoming events through its attention branches:

$$i_d(t+1) = \alpha_d i_d(t) + \sum_{j \in N_d} w_j p_j, \tag{4}$$

  where $\alpha_d = e^{-\frac{1}{\tau_d}}$ represents the decay rate, $w_j$ is the synaptic weight, and $p_j$ is the input event value.

- **Soma Potential Update and Event Generation**: The soma potential is updated based on the integrated EAL attention branch currents:

$$V(t+1) = \beta V(t) + \sum_d g_d i_d(t), \tag{5}$$

  where $\beta = e^{-\frac{1}{\tau_s}}$ is the decay rate of the soma, and $g_d$ is the coupling strength of each attention branch. A event is generated if $V(t)$ exceeds the threshold $V_{\text{th}}$.

- **Event Propagation**: The generated events propagate through the network according to:

$$x(t_{k+1}) = e^{A \Delta t_k} x(t_k) + A^{-1}(e^{A \Delta t_k} - I) B S(t_k), \tag{6}$$

  preserving both spatial and temporal information.

**Methods**

The proposed model is designed to handle sparse, asynchronous event-based inputs effectively while being scalable to high-definition (HD) event streams. It leverages concepts from *Dendrite Heterogeneity Leaky Integrate-and-Fire (DH-LIF)* neurons in the first layer to capture *multi-scale temporal dynamics*, crucial for preserving temporal details inherent in event streams while reducing spatial and computational redundancy. The model then utilizes a series of *event-based state-space convolution* layers, enabling efficient integration of both local and global temporal relationships. The final *readout layer* employs event pooling and a linear transformation to produce a compact and meaningful representation for downstream tasks such as classification or regression. This architecture ensures robustness and scalability, making it suitable for high-resolution inputs.

**Variables and Notations**

To ensure clarity, we provide definitions for all variables and notation used in the equations:

Table 11: Summary of Variables and Notations

| Variable | Definition | Variable | Definition | Variable | Definition |
|---|---|---|---|---|---|
| | | | **Input Representation** | | |
| $x, y$ | Spatial coordinates of the event | $t$ | Timestamp of the event | $p$ | Magnitude or polarity of the event |
| | | | **Event Attention Layer (EAL)** | | |
| $\tau_d$ | EAL timing factor | $i_d(t)$ | EAL Attention current for branch $d$ at $t$ | $\alpha_d$ | Decay rate of EAL $d$, $e^{-\frac{1}{\tau_d}}$ |
| $\mathcal{N}_d$ | Presynaptic inputs to EAL $d$ | $w_j$ | Synaptic weight of presynaptic input $p_j$ | $V(t)$ | Soma membrane potential at $t$ |
| $\beta$ | Soma decay rate, $e^{-\frac{1}{\tau_s}}$ | $g_d$ | Coupling strength of attention branches $d$ | $V_{\text{th}}$ | Threshold potential for event generation |
| | | | **Spatial Pooling Layer** | | |
| $I(x, y, t)$ | Initial event activity | $I_{\text{pooled}}(x', y', t)$ | Pooled event activity | $P(x', y')$ | Pooling window center at $(x', y')$ |
| | | | **FLAME Convolution Layer** | | |
| $x(t)$ | Internal state vector at $t$ | $S(t)$ | Input event train | $A_S$ | EA-HiPPO matrix for inter-event intervals |
| $B, C$ | Input/output coupling matrices | $\Delta t$ | Inter-event interval | $F(\Delta t)$ | EA-HiPPO decay matrix, $e^{-\alpha_{ij}\Delta t}$ |
| $V, \Lambda$ | Components of NPLR decomposition | $P, Q$ | Low-rank matrices, $r \ll N$ | $K(\omega)$ | FFT convolution kernel, $\frac{1}{\omega - \Lambda}$ |
| $\text{FFT}(\cdot)$ | Fast Fourier Transform | $\text{IFFT}(\cdot)$ | Inverse FFT | | |
| | | | **Normalization Layer** | | |
| $x_l$ | Input at layer $l$ | $\mu_l, \sigma_l^2$ | Mean, variance at layer $l$ | $\gamma, \beta$ | Learnable scale and shift parameters |
| | | | **Readout Layer** | | |
| $x_{\text{pooled}, k}$ | Pooled state vector | $W, b$ | Learnable weight matrix and bias | $y$ | Final output, $y = W x_{\text{pooled}} + b$ |

**Overview of the FLAME Model**

FLAME addresses the limitations of conventional event-based neural networks (SNNs) in capturing long-context temporal dependencies while maintaining event-driven efficiency. The FLAME model is composed of the following key components:

---

**Algorithm 2** FLAME Model Processing

---

**Require:** Input event sequence $X = \{(x_i, y_i, t_i, p_i)\}$
  1: **Initialize** model parameters
  2: **Process** input through **Event Attention Layer** (Algorithm 3)
  3: **Apply Spatial Pooling Layer** to reduce spatial dimensions (Algorithm 4)
  4: **Pass** output to **FLAME Convolution Layer** to capture temporal dynamics (Algorithm 5)
  5: **Update** state using **Event-Aware HiPPO** mechanism (Algorithm 5)
  6: **Aggregate** information in the **Readout Layer** for final output (Algorithm 6)
  7: **Output**: Model prediction $y$

---

## C.1  Input Representation

The input to the model is represented as a sequence of events, each defined by the tuple $(x, y, t, p)$, where $(x, y)$ are the spatial coordinates, $t$ is the timestamp, and $p$ represents the magnitude or polarity of the event. These events are streamed asynchronously, reflecting the sparse nature of the data. The model is also designed to handle higher resolutions, allowing scalability to HD event streams. This input representation emphasizes the need for efficient aggregation of both spatial and temporal information while minimizing computational load.

## C.2  Event Attention Layer (EAL)

The model begins by passing the input through the *Event Attention Layer (EAL)*, constructed using DH-LIF neurons as shown in Fig. 1. Each DH-LIF neuron features multiple attention branches, each with a unique timing factor $\tau_d$, enabling the capture of temporal dynamics across a range of timescales, which is essential for accommodating the diverse timescales present in asynchronous event inputs. The dynamics of the attention current $i_d(t)$ are governed by $i_d(t + 1) = \alpha_d i_d(t) + \sum_{j \in \mathcal{N}_d} w_j p_j$, where

$\alpha_d = e^{-\frac{1}{\tau_d}}$ is the decay rate for branch $d$, and $w_j$ represents the synaptic weight associated with presynaptic input $p_j$. The set $\mathcal{N}_d$ represents the presynaptic inputs connected to attention branch $d$, ensuring that each captures temporal features independently, functioning as independent temporal

filters. Unlike a standard CUBA LIF neuron model, which integrates all inputs uniformly at the soma with a single timescale, the event-based attention layer introduces multiple event attention branches, each independently filtering inputs at different temporal scales. This design enables the neuron to selectively process asynchronous inputs and retain information across diverse temporal windows, providing greater flexibility and adaptability.

The currents from each event attention branch are aggregated at the soma, resulting in the membrane potential $V(t+1) = \beta V(t) + \sum_d g_d i_d(t)$, where $\beta = e^{-\frac{1}{\tau_s}}$ represents the soma's decay rate, and $g_d$ represents the coupling strength of attention branch $d$ to the soma. The coupling strength $g_d$ is explicitly separated from the synaptic weights $w_j$ (Equation 4) to allow for independent learning of temporal filtering via $\tau_d$ and $w_j$, and the relative significance of each timescale via $g_d$. This separation empirically improves training stability and prevents performance degradation observed when collapsing the dendritic integration into a flattened structure. A event is generated whenever the membrane potential exceeds a threshold $V_{\text{th}}$, allowing the neuron to selectively fire only when sufficiently excited.

---

**Algorithm 3** Event Attention Layer

---

**Require:** Input events $X = \{(x_i, y_i, t_i, p_i)\}$, timing factors $\{\tau_d\}$, synaptic weights $\{w_j\}$, coupling strengths $\{g_d\}$, threshold $V_{\text{th}}$
 1: **Initialize** currents $i_d(0)$ and membrane potential $V(0)$
 2: **for** each time step $t$ **do**
 3:     **for** each attention branch $d$ **do**
 4:         Compute decay rate: $\alpha_d \leftarrow e^{-\frac{1}{\tau_d}}$
 5:         Update current: $i_d(t+1) \leftarrow \alpha_d i_d(t) + \sum_{j \in \mathcal{N}_d} w_j p_j$
 6:     **end for**
 7:     Compute soma decay rate: $\beta \leftarrow e^{-\frac{1}{\tau_s}}$
 8:     Update membrane potential: $V(t+1) \leftarrow \beta V(t) + \sum_d g_d i_d(t)$
 9:     **if** $V(t+1) > V_{\text{th}}$ **then**
10:         Generate event at time $t+1$
11:         Reset membrane potential: $V(t+1) \leftarrow 0$
12:     **end if**
13: **end for**
14: **Output**: Spatio-temporal features $I(x, y, t)$

---

### C.3 Spatial Pooling Layer

Following the event-based attention layer, a *Spatial Pooling Layer* is introduced to reduce the spatial dimensionality of the resulting output. Given the initial event activity $I(x, y, t)$ at location $(x, y)$, the pooling operation reduces spatial dimensions while preserving temporal resolution:

$$I_{\text{pooled}}(x', y', t) = \max_{(x,y) \in P(x', y')} I(x, y, t)$$

where $P(x', y')$ is a pooling window centered at $(x', y')$. Pooling reduces spatial complexity, simplifying subsequent processing in the network while retaining key features. This is especially useful for HD event streams with extensive spatial information.

---

**Algorithm 4** Spatial Pooling Layer

---

**Require:** Input event activity $I(x, y, t)$ from Event Attention Layer, pooling window $P(x', y')$
 1: **for** each spatial location $(x', y')$ **do**
 2:     **for** each time step $t$ **do**
 3:         Pool activity: $I_{\text{pooled}}(x', y', t) \leftarrow \max_{(x,y) \in P(x', y')} I(x, y, t)$
 4:     **end for**
 5: **end for**
 6: **Output**: Pooled event activity $I_{\text{pooled}}(x', y', t)$

---

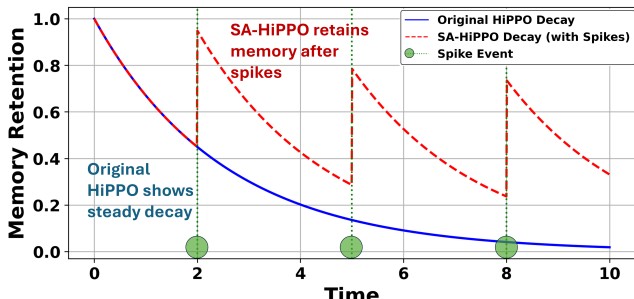

Figure 5: The EA-HiPPO decay is needed to adapt the memory retention dynamically to the irregular timing of events, allowing the system to prioritize recent events while efficiently managing the decay of older information, which enhances stability and responsiveness for event-driven inputs.

## C.4 FLAME Convolution

The **FLAME Convolution Layer** is a critical component of the FLAME model, specifically designed for processing event-based inputs. It captures long-range dependencies and asynchronous dynamics by integrating mechanisms such as the **Event-Aware HiPPO (EA-HiPPO)** framework, **Normal Plus Low-Rank (NPLR) Decomposition**, and **Fast Fourier Transform (FFT) Convolution**. These innovations collectively enable efficient and robust temporal feature extraction.

### Overview and Intuition

Traditional convolutional layers are adept at extracting spatial features but often fail to capture complex temporal dependencies, especially in asynchronous, sparse event-based data. The FLAME Convolution Layer overcomes this limitation by incorporating state-space models that inherently manage temporal dynamics. Leveraging the EA-HiPPO mechanism, the layer dynamically adapts memory retention based on event timings, emphasizing recent events while allowing older information to decay. The use of NPLR Decomposition and FFT-based convolution further enhances computational efficiency, enabling scalability to high-dimensional, long-context temporal data.

**Event-based State-Space Model:** The temporal dynamics of the FLAME Convolution Layer are governed by the **Event-based State-Space Model**:

$$\dot{x}(t) = A_S x(t) + BS(t), \quad y(t) = Cx(t), \tag{7}$$

where:

- $x(t) \in \mathbb{R}^N$ represents the internal state vector,
- $S(t) \in \mathbb{R}^M$ is the input event train, with each component $S_i(t) = \sum_k \delta(t - t_i^k)$, where $\delta(t)$ is the Dirac delta function,
- $A_S \in \mathbb{R}^{N \times N}$ is the **Event-Aware HiPPO** matrix,
- $B \in \mathbb{R}^{N \times M}$ and $C \in \mathbb{R}^{P \times N}$ are the input and output coupling matrices.

This framework ensures that temporal dependencies inherent in event-based data are captured effectively.

**Event-Aware HiPPO Mechanism**: The *Event-Aware HiPPO (EA-HiPPO)* (Fig. 5) mechanism is a core component of the FLAME model, designed to efficiently capture long-term temporal dependencies in the presence of sparse, event-based inputs. The HiPPO (Highly Optimized Polynomial Projection) framework, originally developed to approximate continuous input signals, projects them onto polynomial bases, enabling efficient temporal compression of input history. However, when dealing with event-driven dynamics, where inputs are discrete and irregular, the conventional HiPPO formulation must be adapted to properly address these challenges. The EA-HiPPO adapts the HiPPO framework to efficiently handle discrete, event-driven inputs by introducing a decay matrix $F(\Delta t)$. This matrix adjusts memory retention based on the time elapsed between events ($\Delta t$), ensuring more recent events have a greater influence while older information gradually decays. The Hadamard product with the original HiPPO matrix enables adaptive modulation of memory, making it more

stable and suitable for asynchronous events. In a event-driven scenario, the input signal is represented as a vector of event trains $S(t) \in \mathbb{R}^M$, with each element $S_i(t)$ defined by $S_i(t) = \sum_k \delta(t - t_i^k)$,

where $\delta(t)$ is the Dirac delta function, and $t_i^k$ denotes the time of the $k$-th event for input $i$. Given the irregular and sparse nature of these event-driven inputs, we introduce a *Event-Aware HiPPO (EA-HiPPO)* matrix $A_S$ that extends the dynamics of the standard HiPPO to efficiently process events. The EA-HiPPO matrix $A_S$ modifies the original HiPPO dynamics to adapt to the nature of event-based events by incorporating a decay function that accounts for the time elapsed between successive events. Specifically, the state evolution in the presence of events is modeled by $\dot{x}(t) = A_S x(t) + BS(t)$. The matrix $A_S$ is defined as $A_S = A \circ F(\Delta t)$, where $A \in \mathbb{R}^{N \times N}$ is the original HiPPO matrix, and $F(\Delta t) \in \mathbb{R}^{N \times N}$ is a decay matrix that weights the original HiPPO dynamics based on the inter-event interval $\Delta t$. The operator $\circ$ denotes the element-wise (Hadamard) product. The decay matrix $F(\Delta t)$ is formulated as $F_{ij}(\Delta t) = e^{-\alpha_{ij} \Delta t}$, where $\Delta t = t_j - t_i$ represents the time difference between event $i$ and event $j$, and $\alpha_{ij}$ is a decay parameter that controls how the influence of past events diminishes over time. The exponential decay function ensures that the impact of previous events decreases exponentially, allowing more recent events to have a stronger influence on the current state. This weighting mechanism makes the HiPPO dynamics more adaptable to event-based inputs, capturing both the recency and relevance of events for efficient temporal representation.

The state vector $x(t)$ thus evolves in two distinct modes: continuous evolution between events and instantaneous updates at event times. Between events, the state evolves according to the homogeneous equation $\dot{x}(t) = A_S x(t)$. When a event occurs at time $t_k$, the state is updated as:

$$x(t_{k+1}) = e^{A_S \Delta t_k} x(t_k) + A_S^{-1} \left( e^{A_S \Delta t_k} - I \right) BS(t_k)$$

where $\Delta t_k = t_{k+1} - t_k$ represents the time difference between successive events. To make the state update computationally feasible, the matrix exponential $e^{A_S \Delta t_k}$ is approximated using a truncated Taylor series expansion:

$$e^{A_S \Delta t_k} \approx I + A_S \Delta t_k + \frac{A_S^2 \Delta t_k^2}{2}$$

This first-order or second-order approximation provides a good balance between computational efficiency and accuracy, especially in scenarios with small inter-event intervals.

The EA-HiPPO mechanism effectively extends the temporal memory capabilities of the original HiPPO framework by introducing a event-sensitive adaptation. It ensures that the state vector $x(t)$ retains relevant temporal information while accommodating the asynchronous nature of event inputs. The decay function embedded within $F(\Delta t)$ provides a means to dynamically adjust the influence of past inputs, thereby making the model more responsive to recent events.

**Normal Plus Low-Rank (NPLR) Decomposition**: The **NPLR Decomposition** reduces computational complexity by expressing $A_S$ as:

$$A_S = V \Lambda V^* - P Q^*, \tag{8}$$

where:

- $V \in \mathbb{C}^{N \times N}$ is a unitary matrix,
- $\Lambda \in \mathbb{C}^{N \times N}$ is a diagonal matrix of decay rates,
- $P, Q \in \mathbb{C}^{N \times r}$ are low-rank matrices, with $r \ll N$.

In all experiments, the state dimension is fixed at $N = 64$ and the low-rank perturbation rank is fixed at $r = 8$. This decomposition reduces the complexity of matrix-vector multiplications from $O(N^2)$ to $O(Nr)$, facilitating scalability to large state spaces.

**Fast Fourier Transform (FFT) Convolution**: Long-range temporal dependencies are handled efficiently using FFT-based convolution. The convolution operation is performed as follows:

1. Transform the state vector $x(t)$ and convolution kernel $K(\omega)$ into the frequency domain using FFT.

2. Perform element-wise multiplication in the frequency domain.

3. Apply the inverse FFT (IFFT) to obtain the updated state vector in the time domain.

This approach significantly accelerates the processing of long temporal sequences by leveraging frequency-domain efficiencies. The FLAME Convolution Layer integrates these components to achieve robust spatio-temporal feature extraction:

- **Temporal Dynamics Modeling**: EA-HiPPO captures event timing dependencies while balancing memory retention and decay.
- **Computational Efficiency**: NPLR Decomposition and FFT convolution ensure scalability and rapid processing.
- **Efficient State Management**: The state-space formulation ensures accurate updates for event-based inputs.

### C.4.1 FLAME Convolution Layer

Using all these concepts of EA-Hippo, NPLR Decomposition and FFT Convolution, we introduce *FLAME Convolution (FLAMEConv)* layers, which generalize the event-aware state-space operations into a convolutional framework. These layers are designed to extend the capabilities of FLAME by transforming the temporal memory operations into a convolutional form, thus allowing for more efficient feature extraction in both temporal and spatial domains. The *FLAME Conv* layer incorporates event-based input while retaining the convolutional structure, enabling the model to operate efficiently over high-dimensional data while capturing complex temporal dependencies. The continuous-time state-space dynamics are given by:

$$\frac{d}{dt}x(t) = Ax(t) + Bu(t)$$

where $x(t) \in \mathbb{R}^N$ represents the state vector, $u(t) \in \mathbb{R}^M$ is the input, $A \in \mathbb{R}^{N \times N}$ is the state transition matrix, and $B \in \mathbb{R}^{N \times M}$ is the input coupling matrix. The state evolves based on both the internal dynamics and the influence of incoming events. The Event-Aware dynamics incorporate both decay and event-driven updates

$$\dot{x}(t) = A_{\text{event}}(t)x(t) + B_{\text{event}}(t)u(t), \tag{9}$$

where $A_{\text{event}}(t) = A_{\text{decay}} + A_{\text{timing}}(t)$. The matrix $A_{\text{decay}} = -\frac{1}{\tau_m}I$ models natural decay, while $A_{\text{timing}}(t)$ represents event-driven effects and depends on the inter-event intervals. The model discretizes these dynamics for efficient implementation, using a fixed time step $\Delta t$:

$$x_{k+1} = x_k + \Delta t(A_{\text{event},k}x_k + B_{\text{event},k}u_k) \tag{10}$$

At each event time $t_i$, the state undergoes an instantaneous update $x(t_i^+) = x(t_i^-) + B_{\text{event}}(t_i)$. To improve computational efficiency, the event-based state matrix $A_{\text{event}}$ is decomposed using the *Normal Plus Low-Rank (NPLR) decomposition*: $A_{\text{event}} = V\Lambda V^* - PQ^*$

where $V \in \mathbb{C}^{N \times N}$ is a unitary matrix, $\Lambda \in \mathbb{C}^{N \times N}$ represents the decay, and $P, Q \in \mathbb{C}^{N \times r}$ are low-rank matrices. This reduces the cost of matrix-vector products from $O(N^2)$ to $O(Nr)$, where $r$ is the rank of the low-rank perturbation. The resulting state update rule becomes:

$$x_{k+1} = x_k + \Delta t\left((V\Lambda V^* - PQ^*)x_k + B_{\text{event}}u_k\right)$$

The convolution operation in these layers is realized by transforming recurrent state-space updates into a convolutional form, with the system's impulse response precomputed. Using the *Fast Fourier Transform (FFT)*, the convolution kernel $K(\omega)$ can be efficiently calculated as $K(\omega) = \frac{1}{\omega - \Lambda}$. This transformation allows the model to handle long-range temporal dependencies efficiently, even in high-resolution event-based streams.

**Computational Efficiency:** The layer achieves notable computational advantages:

- **Reduced Complexity**: NPLR Decomposition transforms operations from $O(N^2)$ to $O(Nr)$.

- **Accelerated Convolutions**: FFT convolution rapidly processes long temporal sequences.

- **Parallelization**: FFT operations are well-suited for parallel hardware architectures, enhancing performance.

**Event Generation in FLAME Convolution Layers**: Events in the FLAME model are generated through the interaction of Event attention layer (EAL) and soma compartments in the DH-LIF neurons. These neurons are integral to the Event Attention Layer, which precedes each FLAME convolution layer, ensuring asynchronous and event-driven signal processing.

The EAL branches act as independent temporal filters, accumulating and processing inputs over time:

$$i_d(t+1) = \alpha_d i_d(t) + \sum_{j \in \mathcal{N}_d} w_j p_j,$$

where $\alpha_d = e^{-\frac{1}{\tau_d}}$ is the decay rate determined by the EAL branch's time constant $\tau_d$, $w_j$ is the synaptic weight, and $p_j$ is the presynaptic event.

The soma aggregates these currents, with its membrane potential evolving as:

$$V(t+1) = \beta V(t) + \sum_d g_d i_d(t),$$

where $\beta = e^{-\frac{1}{\tau_s}}$ represents the soma's decay factor, and $g_d$ is the coupling strength of each EAL branch $d$.

A event is produced when the soma's membrane potential $V(t)$ exceeds the threshold $V_{\text{th}}$. After firing, the potential resets, and these events serve as inputs to the next FLAME convolution layer. This mechanism ensures the model maintains its asynchronous event-driven processing nature while enabling precise temporal modeling across layers.

The **FLAME Convolution Layer** combines the strengths of EA-HiPPO, NPLR Decomposition, and FFT Convolution to process asynchronous event-based inputs effectively. This integration enables the model to extract meaningful spatio-temporal features while maintaining computational efficiency and scalability, making it ideal for high-resolution, real-world applications.

## C.5   Normalization and Residual

To maintain stability and ensure efficient learning, *Layer Normalization (LN)* is applied after each event-based SSM convolution layer: $\hat{x}_l = \dfrac{x_l - \mu_l}{\sqrt{\sigma_l^2 + \epsilon}} \cdot \gamma + \beta$, where $\mu_l$ and $\sigma_l^2$ are the mean and variance of activations at layer $l$, respectively, and $\gamma, \beta$ are learnable parameters. Normalization reduces variability in activations, providing stable training regardless of fluctuations in inputs.

Additionally, *residual connections* help propagate information across layers by defining $x_{l+1} = f(x_l) + x_l$, where $f(x_l)$ represents the transformation applied by the event-based convolution at layer $l$. Residual connections prevent vanishing gradients, allow lower-level feature retention, and enhance learning efficiency.

Table 12: Input-Output Descriptions for Each Block in the FLAME Model

| Block | Input | Output |
|---|---|---|
| **Input Representation** | Event events $(x,y,t,p)$: $(x,y)$ (spatial), $t$ (time), $p$ (magnitude/polarity) | Preprocessed events for subsequent layers |
| | Event event stream with spatial and temporal coordinates $(x,y,t,p)$ | Aggregated membrane potential $\mathbf{v}(t)$, capturing spatio-temporal features at multiple timescales. |
| **Event Attention Layer** | *EAL Current Update:* Previous EAL branch current $\mathbf{i}_d(t)$, synaptic weights $\mathbf{w}_j$, and decay factor $\alpha_d$ | Updated EAL branch current $\mathbf{i}_d(t+1) = \alpha_d \mathbf{i}_d(t) + \sum_{j \in \mathcal{N}_d} \mathbf{w}_j p_j$ |
| | *Soma Aggregation:* Inputs from EAL branch currents $\mathbf{i}_d(t+1)$, soma decay factor $\beta$, and coupling strengths $\mathbf{g}_d$ | Aggregated membrane potential $\mathbf{v}(t+1) = \beta \mathbf{v}(t) + \sum_d \mathbf{g}_d \mathbf{i}_d(t)$ |
| | *Event Generation:* | Event output if $\mathbf{v}(t+1) > V_{\text{th}}$, and reset potential $(\mathbf{v}(t+1) \leftarrow 0)$ |
| **Spatial Pooling Layer** | Aggregated events $\mathbf{I}(x,y,t)$ from the Event Attention Layer | Pooled spatio-temporal representation $\mathbf{I}_{\text{pooled}}(x',y',t)$, with reduced spatial dimensions |
| **FLAME Convolution Layer** | Pooled event features $\mathbf{I}_{\text{pooled}}(x',y',t)$ | Processed state $\mathbf{y}(t)$, thresholded to generate events. |
| | *EA-HiPPO:* Event features and inter-event intervals $(\Delta t)$ | Adjusted state-space matrix $\mathbf{A_S}$, incorporating memory retention through a decay matrix |
| | *NPLR Decomposition:* Adjusted state-space matrix $\mathbf{A_S}$ | Decomposed matrix $\mathbf{A_S} = \mathbf{V}\mathbf{\Lambda}\mathbf{V}^* - \mathbf{P}\mathbf{Q}^*$, reducing computational complexity |
| | *Matrix Exponential Approximation:* Decomposed state-space matrix $\mathbf{A_S}$, time step $\Delta t_k$ | Approximated exponential $e^{\mathbf{A_S}\Delta t_k}$ for efficient state updates |
| | *FFT Convolution:* State vector $\mathbf{x}(t_k)$ and precomputed impulse response $\mathbf{K}(\omega)$ | Updated state vector $\mathbf{x}(t_{k+1})$ after efficient frequency-domain convolution |
| **Layer Normalization** | Intermediate activations $\mathbf{x}_l$ from the FLAME Convolution Layer | Normalized activations $\hat{\mathbf{x}}_l$, ensuring stable training by reducing variability in activations |
| **Readout Layer** | Normalized features $\hat{\mathbf{x}}_l$ | Final output $\mathbf{y}$, generated via event pooling and a linear transformation |

---

**Algorithm 5** FLAME Convolution Layer

---

**Require:** Event train input $S(t)$, HiPPO base matrix $\mathbf{A}$, input coupling matrix $\mathbf{B}$, output coupling matrix $\mathbf{C}$, decay function $\mathbf{F}(\Delta t)$, time step $\Delta t$, low-rank matrices $\mathbf{P}, \mathbf{Q}$, total time $T$, rank $r$, state space dimension $N$, FFT convolution kernel $\mathbf{K}(\omega)$, threshold potential $V_{\text{th}}$
**Ensure:** Output event map $Y_{\text{event}}(t)$
    **Initialization**
1: Initialize state vector $\mathbf{x} \leftarrow 0$                                            *(N-dimensional state vector)*
2: Initialize output $Y_{\text{event}} \leftarrow []$                                 *(Empty list to store event outputs)*
    **Precomputations**
3: Compute event-aware HiPPO matrix: $\mathbf{A}_{\text{event}} \leftarrow \mathbf{A} \circ \mathbf{F}(\Delta t)$      *(Hadamard product with decay function)*
4: Perform eigendecomposition: $\mathbf{V}, \mathbf{\Lambda} \leftarrow \text{eig}(\mathbf{A}_{\text{event}})$
5: Decompose using NPLR: $\mathbf{A}_{\text{NPLR}} \leftarrow \mathbf{V}\mathbf{\Lambda}\mathbf{V}^* - \mathbf{P}\mathbf{Q}^*$
6: **for** $t = 1$ **to** $T$ **do**
    **Event-Driven Dynamics**
7:     **if** $S(t)$ contains events **then**
8:         Compute time difference: $\Delta t_k = t_{k+1} - t_k$
9:         Approximate matrix exponential:

$$e^{\mathbf{A}_{\text{event}}\Delta t_k} \approx \mathbf{I} + \mathbf{A}_{\text{event}}\Delta t_k + \frac{(\mathbf{A}_{\text{event}})^2(\Delta t_k)^2}{2}$$

10:         Update state vector:

$$\mathbf{x}(t_{k+1}) \leftarrow \mathbf{x}(t_k) + \Delta t_k \left((\mathbf{V}\mathbf{\Lambda}\mathbf{V}^* - \mathbf{P}\mathbf{Q}^*)\mathbf{x}(t_k) + \mathbf{B}S(t_k)\right)$$

11:     **else**
12:         Update state for continuous dynamics: $\mathbf{x} \leftarrow e^{\mathbf{A}_{\text{event}}\Delta t}\mathbf{x}$
13:     **end if**
    **FFT-Based Convolution for Temporal Dependencies**
14:     Transform state and kernel to frequency domain:

$$\mathbf{X}_{\text{freq}} \leftarrow \text{FFT}(\mathbf{x}), \quad \mathbf{K}_{\text{freq}} \leftarrow \text{FFT}(\mathbf{K}(\omega))$$

15:     Perform element-wise multiplication in frequency domain:

$$\mathbf{Y}_{\text{freq}} \leftarrow \mathbf{X}_{\text{freq}} \cdot \mathbf{K}_{\text{freq}}$$

16:     Transform back to time domain:

$$\mathbf{x}(t_{k+1}) \leftarrow \text{IFFT}(\mathbf{Y}_{\text{freq}})$$

17:     Compute continuous output: $y_t \leftarrow \mathbf{C} \cdot \mathbf{x}(t)$
18:     Threshold the output to generate events:

$$y_{\text{event}}(t) \leftarrow \mathbb{I}(y_t > V_{\text{th}})$$

19:     Append $y_{\text{event}}(t)$ to $Y_{\text{event}}$
20: **end for**
    **Output:** $Y_{\text{event}}$, the final event map

---

## C.6   Readout Layer

The readout layer is inspired by the *Event-SSM* architecture and employs an *event-pooling mechanism* to subsample the temporal sequence length. The pooled output is computed as $x_{\text{pooled},k} = \frac{1}{p}\sum_{i=kp}^{(k+1)p-1} x_i$, where $p$ is the pooling factor. This operation ensures only the most relevant temporal features are retained, reducing computational burden while preserving key information. The resulting pooled sequence is passed through a linear transformation as $y = Wx_{\text{pooled}} + b$ where $W$ and $b$ are learnable parameters. The combination of event pooling and linear transformation provides

---

**Algorithm 6** Readout Layer

---

**Require:** State vectors $\{x(t)\}$, pooling factor $p$, weights $W$, bias $b$
 1: **for** each pooled time step $k$ **do**
 2:     Compute pooled state:

$$x_{\text{pooled},k} \leftarrow \frac{1}{p} \sum_{i=kp}^{(k+1)p-1} x(t_i)$$

 3: **end for**
 4: Compute final output:

$$y \leftarrow W x_{\text{pooled}} + b$$

 5: **Output**: Model prediction $y$

---

an efficient means for deriving a final representation suitable for downstream tasks, maintaining scalability even with longer event sequences.

## C.7   FLOPs Calculation Methodology

The estimation of Floating Point Operations (FLOPs) per event in the FLAME model considers the computational costs associated with the primary components involved in processing each event. The methodology is detailed below:

1. **Event-Driven EAL Integration:** Each incoming event can trigger current updates across multiple EAL branches. The FLOPs for this stage are approximated as:

$$\text{FLOPs}_{\text{EAL}} \approx (\text{No. of branches} \times \text{No. of inputs per branch} \times 2)$$

    This accounts for one addition and one multiplication operation per active synapse connected to the EAL branches influenced by the event. Let $d$ represent the complexity related to EAL operations, so $\text{FLOPs}_{\text{EAL}} = O(d)$.

2. **EA-HiPPO State Update:** The Event-Aware HiPPO (EA-HiPPO) mechanism updates its state upon receiving input. When using a 2nd-order matrix exponential approximation and NPLR decomposition, each update involves:

    - Approximately $2 \times (N \times r)$ FLOPs for the Normal-Plus-Low-Rank (NPLR) matrix-vector products, where $N$ is the state dimension and $r$ is the rank of the low-rank component.
    - $O(N \log N)$ FLOPs for the Fast Fourier Transform (FFT) and Inverse FFT (IFFT) operations used in the convolutional implementation of the state space model.

    Thus, the FLOPs for the EA-HiPPO component can be expressed as:

$$\text{FLOPs}_{\text{EA-HiPPO}} \approx 2Nr + c \cdot N \log N$$

    where $c$ is a small constant factor associated with the FFT/IFFT computations.

3. **Total Per-Event FLOPs:** The total computational cost per event is the sum of the FLOPs from the EAL integration and the EA-HiPPO state update:

$$\text{FLOPs}_{\text{per\_event}} = \text{FLOPs}_{\text{EAL}} + \text{FLOPs}_{\text{EA-HiPPO}}$$

    Substituting the approximations, we get:

$$\text{FLOPs}_{\text{per\_event}} \approx O(d) + 2Nr + O(N \log N)$$

This methodology provides an estimate of the computational load incurred for each event processed by the FLAME architecture, highlighting the contributions of its main computational blocks. The NPLR decomposition and the efficiency of FFT-based convolutions are key to maintaining manageable FLOP counts, especially in comparison to dense matrix operations.

# NeurIPS Paper Checklist

1. **Claims**

   Question: Do the main claims made in the abstract and introduction accurately reflect the paper's contributions and scope?

   Answer: [Yes]

   Justification: The final step for preparing your paper is to complete the Claims section of the checklist with a concise, evidence-based justification. The justification must affirm that the paper's main contributions are accurately represented and experimentally supported. You should state that the claims regarding the Event Attention Layer (EAL) and Event-Aware HiPPO (EA-HiPPO) are backed by ablations showing significant performance drops when they're removed (Fig. 2a). Furthermore, emphasize that the claim of computational efficiency is proven by achieving competitive accuracy on high-resolution data (CeleX-HAR) at $\approx 90\times$ lower GFLOPs than the leading voxel-based baseline, and by the technical contribution of NPLR decomposition.

   Guidelines:

   - The answer NA means that the abstract and introduction do not include the claims made in the paper.
   - The abstract and/or introduction should clearly state the claims made, including the contributions made in the paper and important assumptions and limitations. A No or NA answer to this question will not be perceived well by the reviewers.
   - The claims made should match theoretical and experimental results, and reflect how much the results can be expected to generalize to other settings.
   - It is fine to include aspirational goals as motivation as long as it is clear that these goals are not attained by the paper.

2. **Limitations**

   Question: Does the paper discuss the limitations of the work performed by the authors?

   Answer: [Yes]

   Justification: The paper explicitly discusses several limitations in Section 6, detailing areas where the current framework's scope is restricted. The authors acknowledge that, despite achieving high throughput on modern GPUs, the current implementation does not meet real-time performance constraints on CPU-only systems and lacks mechanisms for on-device continual learning, as it is designed for static offline training. Furthermore, the model is inherently sensitive to sensor noise —a known trade-off for event-driven models—and its sophisticated state-space core is not directly translatable for end-to-end native execution on existing neuromorphic hardware platforms.

   Guidelines:

   - The answer NA means that the paper has no limitation while the answer No means that the paper has limitations, but those are not discussed in the paper.
   - The authors are encouraged to create a separate "Limitations" section in their paper.
   - The paper should point out any strong assumptions and how robust the results are to violations of these assumptions (e.g., independence assumptions, noiseless settings, model well-specification, asymptotic approximations only holding locally). The authors should reflect on how these assumptions might be violated in practice and what the implications would be.
   - The authors should reflect on the scope of the claims made, e.g., if the approach was only tested on a few datasets or with a few runs. In general, empirical results often depend on implicit assumptions, which should be articulated.
   - The authors should reflect on the factors that influence the performance of the approach. For example, a facial recognition algorithm may perform poorly when image resolution is low or images are taken in low lighting. Or a speech-to-text system might not be used reliably to provide closed captions for online lectures because it fails to handle technical jargon.
   - The authors should discuss the computational efficiency of the proposed algorithms and how they scale with dataset size.

- If applicable, the authors should discuss possible limitations of their approach to address problems of privacy and fairness.
- While the authors might fear that complete honesty about limitations might be used by reviewers as grounds for rejection, a worse outcome might be that reviewers discover limitations that aren't acknowledged in the paper. The authors should use their best judgment and recognize that individual actions in favor of transparency play an important role in developing norms that preserve the integrity of the community. Reviewers will be specifically instructed to not penalize honesty concerning limitations.

3. **Theory assumptions and proofs**

Question: For each theoretical result, does the paper provide the full set of assumptions and a complete (and correct) proof?

Answer: [Yes]

Justification: The paper includes full theoretical analysis, with all proofs provided in the Supplementary Material (Appendix A). Assumptions for each result are clearly stated or referenced: for instance, **Theorem 1 and Theorem 2** explicitly state the requirements for **Hurwitz stability** ($\mathrm{Re}(\lambda_i) < 0$) of the adaptive state matrix $\mathbf{A}_S(\Delta t)$ and the boundedness of inputs ($\|\mathbf{E}_{\text{flat}}(t)\| \leq E_\infty$), ensuring system stability and bounded memory retention. All theorems and lemmas are numbered and correctly cross-referenced.

Guidelines:

- The answer NA means that the paper does not include theoretical results.
- All the theorems, formulas, and proofs in the paper should be numbered and cross-referenced.
- All assumptions should be clearly stated or referenced in the statement of any theorems.
- The proofs can either appear in the main paper or the supplemental material, but if they appear in the supplemental material, the authors are encouraged to provide a short proof sketch to provide intuition.
- Inversely, any informal proof provided in the core of the paper should be complemented by formal proofs provided in appendix or supplemental material.
- Theorems and Lemmas that the proof relies upon should be properly referenced.

4. **Experimental result reproducibility**

Question: Does the paper fully disclose all the information needed to reproduce the main experimental results of the paper to the extent that it affects the main claims and/or conclusions of the paper (regardless of whether the code and data are provided or not)?

Answer: [Yes]

Justification: The main claims accurately reflect the paper's core contributions in efficient, asynchronous processing, backed by strong experimental results. The core architectural elements—the Event Attention Layer (EAL) and Event-Aware HiPPO (EA-HiPPO)—are verified by ablation studies showing their removal leads to significant performance drops across tasks (Fig. 2a). The claim of superior computational efficiency is definitively supported by achieving competitive accuracy on the high-resolution CeleX-HAR dataset at $\approx 90\times$ lower GFLOPs than the voxel-based EventMamba baseline. This efficiency is enabled by the Normal-Plus-Low-Rank (NPLR) decomposition, which drastically reduces complexity and memory footprint.

Guidelines:

- The answer NA means that the paper does not include experiments.
- If the paper includes experiments, a No answer to this question will not be perceived well by the reviewers: Making the paper reproducible is important, regardless of whether the code and data are provided or not.
- If the contribution is a dataset and/or model, the authors should describe the steps taken to make their results reproducible or verifiable.
- Depending on the contribution, reproducibility can be accomplished in various ways. For example, if the contribution is a novel architecture, describing the architecture fully might suffice, or if the contribution is a specific model and empirical evaluation, it may

be necessary to either make it possible for others to replicate the model with the same dataset, or provide access to the model. In general. releasing code and data is often one good way to accomplish this, but reproducibility can also be provided via detailed instructions for how to replicate the results, access to a hosted model (e.g., in the case of a large language model), releasing of a model checkpoint, or other means that are appropriate to the research performed.

- While NeurIPS does not require releasing code, the conference does require all submissions to provide some reasonable avenue for reproducibility, which may depend on the nature of the contribution. For example

    (a) If the contribution is primarily a new algorithm, the paper should make it clear how to reproduce that algorithm.

    (b) If the contribution is primarily a new model architecture, the paper should describe the architecture clearly and fully.

    (c) If the contribution is a new model (e.g., a large language model), then there should either be a way to access this model for reproducing the results or a way to reproduce the model (e.g., with an open-source dataset or instructions for how to construct the dataset).

    (d) We recognize that reproducibility may be tricky in some cases, in which case authors are welcome to describe the particular way they provide for reproducibility. In the case of closed-source models, it may be that access to the model is limited in some way (e.g., to registered users), but it should be possible for other researchers to have some path to reproducing or verifying the results.

5. **Open access to data and code**

   Question: Does the paper provide open access to the data and code, with sufficient instructions to faithfully reproduce the main experimental results, as described in supplemental material?

   Answer: [Yes]

   Justification: The authors ensure full open access to faithfully reproduce the main experimental results. The paper confirms that the full codebase used for all experiments—including the complete Event-Aware HiPPO module with NPLR and FFT support, the full EAL implementation, spatial pooling routines, and all necessary training/evaluation scripts is available in the author's github. Since all required datasets (e.g., DVS Gesture, CeleX-HAR, SHD/SSC) are publicly available, this commitment, coupled with the detailed architectural and hyperparameter descriptions in the paper and appendix, provides a complete and verifiable path to reproduction.

   Guidelines:

   - The answer NA means that paper does not include experiments requiring code.
   - Please see the NeurIPS code and data submission guidelines (`https://nips.cc/public/guides/CodeSubmissionPolicy`) for more details.
   - While we encourage the release of code and data, we understand that this might not be possible, so "No" is an acceptable answer. Papers cannot be rejected simply for not including code, unless this is central to the contribution (e.g., for a new open-source benchmark).
   - The instructions should contain the exact command and environment needed to run to reproduce the results. See the NeurIPS code and data submission guidelines (`https://nips.cc/public/guides/CodeSubmissionPolicy`) for more details.
   - The authors should provide instructions on data access and preparation, including how to access the raw data, preprocessed data, intermediate data, and generated data, etc.
   - The authors should provide scripts to reproduce all experimental results for the new proposed method and baselines. If only a subset of experiments are reproducible, they should state which ones are omitted from the script and why.
   - At submission time, to preserve anonymity, the authors should release anonymized versions (if applicable).
   - Providing as much information as possible in supplemental material (appended to the paper) is recommended, but including URLs to data and code is permitted.

6. **Experimental setting/details**

   Question: Does the paper specify all the training and test details (e.g., data splits, hyper-parameters, how they were chosen, type of optimizer, etc.) necessary to understand the results?

   Answer: [Yes]

   Justification: The paper specifies the majority of the details necessary to understand the results, primarily through the appendix. All model variants (Tiny, Small, Normal) and their architectural details (e.g., channel counts, readout layers) are provided. Key experimental methodologies are detailed, such as event-by-event processing and ablation setups. Furthermore, the paper promises explicit details on critical aspects for reproduction, including the specific optimizer, sequence lengths, EAL branch counts, and state/rank choices, will be included in the final Appendix.

   Guidelines:

   - The answer NA means that the paper does not include experiments.
   - The experimental setting should be presented in the core of the paper to a level of detail that is necessary to appreciate the results and make sense of them.
   - The full details can be provided either with the code, in appendix, or as supplemental material.

7. **Experiment statistical significance**

   Question: Does the paper report error bars suitably and correctly defined or other appropriate information about the statistical significance of the experiments?

   Answer: [Yes]

   Justification: The paper provides limited information on statistical significance. Results are primarily reported as single point estimates (e.g., accuracy percentage and GFLOPs) without accompanying error bars, confidence intervals, or information on the variability factors (e.g., initialization or multiple runs). While the performance figures are clear, the lack of variability metrics means the statistical significance of the differences between the proposed method and baselines is not formally quantified.

   Guidelines:

   - The answer NA means that the paper does not include experiments.
   - The authors should answer "Yes" if the results are accompanied by error bars, confidence intervals, or statistical significance tests, at least for the experiments that support the main claims of the paper.
   - The factors of variability that the error bars are capturing should be clearly stated (for example, train/test split, initialization, random drawing of some parameter, or overall run with given experimental conditions).
   - The method for calculating the error bars should be explained (closed form formula, call to a library function, bootstrap, etc.)
   - The assumptions made should be given (e.g., Normally distributed errors).
   - It should be clear whether the error bar is the standard deviation or the standard error of the mean.
   - It is OK to report 1-sigma error bars, but one should state it. The authors should preferably report a 2-sigma error bar than state that they have a 96% CI, if the hypothesis of Normality of errors is not verified.
   - For asymmetric distributions, the authors should be careful not to show in tables or figures symmetric error bars that would yield results that are out of range (e.g. negative error rates).
   - If error bars are reported in tables or plots, The authors should explain in the text how they were calculated and reference the corresponding figures or tables in the text.

8. **Experiments compute resources**

   Question: For each experiment, does the paper provide sufficient information on the computer resources (type of compute workers, memory, time of execution) needed to reproduce the experiments?

Answer: [Yes]

Justification: Sufficient information is provided to quantify the compute resources required. The main training and inference runs are stated to be conducted on NVIDIA A100 GPUs. Detailed efficiency metrics (GFLOPs, parameters, and inference latency in milliseconds) are provided for all major experiments (Figs. 2, 3, Table 2). Furthermore, Appendix B.6 provides a hardware deployment and efficiency profile for the FLAME-Tiny model across a range of compute workers, giving explicit latency figures.

Guidelines:

- The answer NA means that the paper does not include experiments.
- The paper should indicate the type of compute workers CPU or GPU, internal cluster, or cloud provider, including relevant memory and storage.
- The paper should provide the amount of compute required for each of the individual experimental runs as well as estimate the total compute.
- The paper should disclose whether the full research project required more compute than the experiments reported in the paper (e.g., preliminary or failed experiments that didn't make it into the paper).

9. **Code of ethics**

Question: Does the research conducted in the paper conform, in every respect, with the NeurIPS Code of Ethics https://neurips.cc/public/EthicsGuidelines?

Answer: [Yes]

Justification: The research conforms to the NeurIPS Code of Ethics. This work is fundamental research focused on developing efficient neural network architectures for event-based vision using public, established benchmark datasets (e.g., DVS Gesture, CeleX-HAR). It does not involve human subjects, collect new private or sensitive data, or present immediate dual-use concerns that would necessitate special consideration.

Guidelines:

- The answer NA means that the authors have not reviewed the NeurIPS Code of Ethics.
- If the authors answer No, they should explain the special circumstances that require a deviation from the Code of Ethics.
- The authors should make sure to preserve anonymity (e.g., if there is a special consideration due to laws or regulations in their jurisdiction).

10. **Broader impacts**

Question: Does the paper discuss both potential positive societal impacts and negative societal impacts of the work performed?

Answer: [Yes]

Justification: The research conforms to the NeurIPS Code of Ethics. This work is fundamental research focused on developing efficient neural network architectures for event-based vision using public, established benchmark datasets (e.g., DVS Gesture, CeleX-HAR). It does not involve human subjects, collect new private or sensitive data, or present immediate dual-use concerns that would necessitate special consideration.

Guidelines:

- The answer NA means that there is no societal impact of the work performed.
- If the authors answer NA or No, they should explain why their work has no societal impact or why the paper does not address societal impact.
- Examples of negative societal impacts include potential malicious or unintended uses (e.g., disinformation, generating fake profiles, surveillance), fairness considerations (e.g., deployment of technologies that could make decisions that unfairly impact specific groups), privacy considerations, and security considerations.
- The conference expects that many papers will be foundational research and not tied to particular applications, let alone deployments. However, if there is a direct path to any negative applications, the authors should point it out. For example, it is legitimate to point out that an improvement in the quality of generative models could be used to

generate deepfakes for disinformation. On the other hand, it is not needed to point out that a generic algorithm for optimizing neural networks could enable people to train models that generate Deepfakes faster.

- The authors should consider possible harms that could arise when the technology is being used as intended and functioning correctly, harms that could arise when the technology is being used as intended but gives incorrect results, and harms following from (intentional or unintentional) misuse of the technology.
- If there are negative societal impacts, the authors could also discuss possible mitigation strategies (e.g., gated release of models, providing defenses in addition to attacks, mechanisms for monitoring misuse, mechanisms to monitor how a system learns from feedback over time, improving the efficiency and accessibility of ML).

11. **Safeguards**

Question: Does the paper describe safeguards that have been put in place for responsible release of data or models that have a high risk for misuse (e.g., pretrained language models, image generators, or scraped datasets)?

Answer: [NA]

Justification: The paper poses no such risks. This work is foundational research developing a novel neural network architecture. It does not involve the release of large pretrained generative models (like LLMs or image generators) or the creation/curation of any scraped or sensitive datasets. All data used consists of established, publicly available benchmark datasets in the event-based vision community.

Guidelines:

- The answer NA means that the paper poses no such risks.
- Released models that have a high risk for misuse or dual-use should be released with necessary safeguards to allow for controlled use of the model, for example by requiring that users adhere to usage guidelines or restrictions to access the model or implementing safety filters.
- Datasets that have been scraped from the Internet could pose safety risks. The authors should describe how they avoided releasing unsafe images.
- We recognize that providing effective safeguards is challenging, and many papers do not require this, but we encourage authors to take this into account and make a best faith effort.

12. **Licenses for existing assets**

Question: Are the creators or original owners of assets (e.g., code, data, models), used in the paper, properly credited and are the license and terms of use explicitly mentioned and properly respected?

Answer: [Yes]

Justification: The paper respects all licensing requirements for existing assets. The work exclusively uses publicly available, established benchmark datasets (e.g., DVS Gesture, CeleX-HAR, SHD/SSC, CIFAR10-DVS, N-Caltech101), and the respective original papers for all datasets are properly cited. As these are standard, openly cited research assets, this constitutes proper crediting and adherence to community norms.

Guidelines:

- The answer NA means that the paper does not use existing assets.
- The authors should cite the original paper that produced the code package or dataset.
- The authors should state which version of the asset is used and, if possible, include a URL.
- The name of the license (e.g., CC-BY 4.0) should be included for each asset.
- For scraped data from a particular source (e.g., website), the copyright and terms of service of that source should be provided.
- If assets are released, the license, copyright information, and terms of use in the package should be provided. For popular datasets, `paperswithcode.com/datasets` has curated licenses for some datasets. Their licensing guide can help determine the license of a dataset.

- For existing datasets that are re-packaged, both the original license and the license of the derived asset (if it has changed) should be provided.
- If this information is not available online, the authors are encouraged to reach out to the asset's creators.

13. **New assets**

Question: Are new assets introduced in the paper well documented and is the documentation provided alongside the assets?

Answer: [Yes]

Justification: The paper introduces a novel architectural framework, FLAME, which constitutes a new asset. This asset is well-documented via: 1) A full architectural description in Section 3, including detailed mathematical modeling of the Event Attention Layer and the Event-Aware HiPPO dynamics. 2) Explicit commitment to releasing the complete codebase, including NPLR logic and training scripts, alongside the final paper, which will ensure complete documentation is provided with the assets upon release. No new datasets or models posing high safety risks are introduced.

Guidelines:

- The answer NA means that the paper does not release new assets.
- Researchers should communicate the details of the dataset/code/model as part of their submissions via structured templates. This includes details about training, license, limitations, etc.
- The paper should discuss whether and how consent was obtained from people whose asset is used.
- At submission time, remember to anonymize your assets (if applicable). You can either create an anonymized URL or include an anonymized zip file.

14. **Crowdsourcing and research with human subjects**

Question: For crowdsourcing experiments and research with human subjects, does the paper include the full text of instructions given to participants and screenshots, if applicable, as well as details about compensation (if any)?

Answer: [NA]

Justification: The research does not involve crowdsourcing experiments or primary research with human subjects. The experiments rely exclusively on publicly available, established benchmark datasets that were collected previously by other researchers.

Guidelines:

- The answer NA means that the paper does not involve crowdsourcing nor research with human subjects.
- Including this information in the supplemental material is fine, but if the main contribution of the paper involves human subjects, then as much detail as possible should be included in the main paper.
- According to the NeurIPS Code of Ethics, workers involved in data collection, curation, or other labor should be paid at least the minimum wage in the country of the data collector.

15. **Institutional review board (IRB) approvals or equivalent for research with human subjects**

Question: Does the paper describe potential risks incurred by study participants, whether such risks were disclosed to the subjects, and whether Institutional Review Board (IRB) approvals (or an equivalent approval/review based on the requirements of your country or institution) were obtained?

Answer: [NA]

Justification: The paper does not involve crowdsourcing or primary research with human subjects. The experiments rely exclusively on publicly available, established benchmark datasets that were collected previously by other researchers, making IRB approval non-applicable.

Guidelines:

- The answer NA means that the paper does not involve crowdsourcing nor research with human subjects.
- Depending on the country in which research is conducted, IRB approval (or equivalent) may be required for any human subjects research. If you obtained IRB approval, you should clearly state this in the paper.
- We recognize that the procedures for this may vary significantly between institutions and locations, and we expect authors to adhere to the NeurIPS Code of Ethics and the guidelines for their institution.
- For initial submissions, do not include any information that would break anonymity (if applicable), such as the institution conducting the review.

16. **Declaration of LLM usage**

Question: Does the paper describe the usage of LLMs if it is an important, original, or non-standard component of the core methods in this research? Note that if the LLM is used only for writing, editing, or formatting purposes and does not impact the core methodology, scientific rigorousness, or originality of the research, declaration is not required.

Answer: [Yes]

Justification: The paper does include a declaration of LLM usage, as acknowledged in the submission's metadata. The usage was primarily for editing, formatting, clarifying technical concepts, and generating/filtering data for the submission, not as an integral part of the core architectural method, scientific rigorousness, or originality of the research.

Guidelines:

- The answer NA means that the core method development in this research does not involve LLMs as any important, original, or non-standard components.
- Please refer to our LLM policy (`https://neurips.cc/Conferences/2025/LLM`) for what should or should not be described.

