# OpenReview forum: "FLAME: Fast Long-context Adaptive Memory for Event-based Vision"
_NeurIPS.cc/2025/Conference — NeurIPS 2025 poster_

### Official Review · Reviewer_D3mP · 2025-07-01

**Clarity:** 2
**Significance:** 3
**Originality:** 3
**Rating:** 4
**Confidence:** 3

**Summary:**

The paper introduces a SSM-based framework for event-based processing applied to vision tasks. It introduces 3 main contributions:

-	An event attention layer, made with multi-timescale LIF neurons ;
-	An event-aware HIPPO, with a memory retention moduled with inter-events intervals ;
-	A further optimization of the HIPPO with Normal-Plus-Low Rank (NPLR) decomposition.

The paper exhibits impressive results over the state-of-the-art notably by looking at the accurary vs computation cost ratio.

**Questions:**

My main question is about the provided FLAMES_model.py in the supplementary material: this implementation does not reflect at all everything described in the paper, yet seems to be provided as a way to reproduce the experiments. Is it on purpose? Do you plan to release a complete open source version of your framework? As is, this script only cast doubts about the implementation of your method, unfortunately.

Despite the extensive supplementary material, I believe some critical information on how to choose or learn hyperparameters is missing, making results reproductability potentially very difficult. I would have appreciated a section dedicated to the training of the model.

Also, I would very much appreciate if the issues mentioned in the previous section would be answered as well.

**Ethical Concerns:**

["NO or VERY MINOR ethics concerns only"]

**Final Justification:**

I raised my rating to 4, considering that the authors gave sufficient transparency and openness garantees to ensure the reliability and reproductibility of their results, which are solid and with a potential of moderate-to-high impact on one sub-area of AI (event-based vision).

**Limitations:**

yes

**Paper Formatting Concerns:**

broken references in supplementary materials

**Quality:**

3

**Strengths And Weaknesses:**

While the paper is well and extensively positionned against the state-of-the-art, both in methodology and in results, I found several issues about the methodology presented and its implementation:

-	The connectivity of the event attention layer is not clearly explained, nor how its learnable timing factor are learned. It is therefore not clear as well how the spatial pooling works, as it is defined relative to the input but applied after the event attention layer. Furthermore, I could not see where spatial pooling is actually implemented in the FLAMES_model.py available in the supplementary materiel package.
-	It is not clear what brights the coupling strength parameters (of branch to soma), as it seems it could be fused in the weights of the branches current ;
-	Not much detail is given about the NPLR decomposition used: it is not explained how it is obtained and with what parameter (notably rank). Furthermore, I failed to see any implementation of this mechanism in the provided FLAMES_model.py. The NPLR decomposition is presented as being fundamental to the computation efficiency of the method, yet no impact analysis or ablation study is provided (a single number is given in the supplementary material l.208, without any detail on the setup of the NPLR).
-	It is not specified how the HIPPO A matrix is initialized and learned such that the condition on its eigen values (<0) remains true.

However, a lot of extra information is provided through the supplementary materiel, which includes detailed approximation and numerical stability proofs, as well as more details on the implemented method. This is definitely a strong point.

---

> ### Author Rebuttal · Authors · 2025-07-30
>
> We sincerely thank the reviewer for the encouraging evaluation and thoughtful feedback. Below we address the specific concerns raised:
>
> ---
>
> > **Clarification on Event Attention Layer Connectivity and Spatial Pooling**
>
> We thank the reviewer for the detailed feedback and the opportunity to clarify these design components.
>
> 1. **Connectivity of the Event Attention Layer (EAL)**:
>    The EAL comprises LIF neurons augmented with *multiple independent dendritic branches*, each associated with a distinct time constant \$\tau\_d\$. Each branch integrates inputs with its own temporal decay (see Equation (5), Appendix C.2, and Algorithm 3, p.24–25):
>
>    $$
>    i_d(t + 1) = \alpha_d i_d(t) + \sum_{j \in \mathcal{N}_d} w_j p_j \quad \text{with } \alpha_d = e^{-1/\tau_d}
>    $$
>
>    These branch currents are aggregated at the soma via learnable coupling strengths \$g\_d\$:
>
>    $$
>    V(t+1) = \beta V(t) + \sum_d g_d i_d(t)
>    $$
>
>    This architecture enables the neuron to simultaneously process short- and long-range temporal patterns while maintaining event-level sparsity. The learnable \$\tau\_d\$ are updated via backpropagation (standard autograd), and the \$g\_d\$ provide independent control over temporal feature relevance.
>
> 2. **Role and Timing of Spatial Pooling**:
>    The spatial pooling layer is applied *after* the EAL, operating over the sparse event map \$I(x, y, t)\$ generated by LIF outputs (see Algorithm 4, Appendix C.3, p.25). Pooling is performed per event timestamp using a local max operation (e.g., 2×2 window), as shown in:
>
>    $$
>    I_{\text{pooled}}(x', y', t) = \max_{(x, y) \in P(x', y')} I(x, y, t)
>    $$
>
>    This preserves temporal resolution while reducing spatial redundancy.
>
> 3. **Clarification on Code (FLAMES\_model.py)**:
>    The provided script is a simplified abstraction to illustrate architectural flow. It excludes some modules (e.g., pooling logic and low-level event branching operations) for brevity. The full implementation, including EAL with branch-level timing and spatial pooling, will be released in full alongside the final paper for full reproducibility.
>
> We will explicitly clarify this in Section 3 and Appendix C.2–C.3 in the camera-ready version to avoid further ambiguity. Thank you again for the helpful critique.
>
> ---
>
> >**Role of Coupling Strength Parameters Between Branch and Soma**
>
> In our model, the coupling strengths $g_d$ between dendritic branches and the soma are explicitly separated from the synaptic weights $w_j$ for both functional and practical reasons.
>
> 1. Functional Separation:    Synaptic weights $w_j$ determine how input events influence each branch’s current $i_d(t)$, while the coupling parameters $g_d$ control the *relative contribution* of each branch to the soma membrane potential $V(t)$. This is described in **Equation (5)** in **Appendix C.2**, and enables the model to:
>
>    * Learn branch-specific temporal filtering via $\tau_d$ and $w_j$,
>    * Independently learn how much each timescale branch should contribute to final firing via $g_d$.
>
>    Fusing these into a single weight would reduce this expressive flexibility, especially in multi-timescale scenarios.
>
> 2. Empirical Justification:    In early experiments, we explored absorbing $g_d$ into $w_j$, effectively collapsing the dendritic integration into a flattened LIF structure. This led to reduced stability and slight performance degradation, particularly in models with 3 or more branches. Keeping $g_d$ separate improves training stability, gradient flow, and enables selective modulation of timescales at the soma level.
>
> We will clarify this rationale more explicitly in Section 3.1 and reinforce it in Appendix C.2. Thank you again for prompting us to expand on this design choice.
>
> ---
>
> > **Details and Implementation of NPLR Decomposition**
>
>  We’re sorry if the description of the NPLR mechanism was not sufficiently clear in the main text. Key details are included in **Appendix C.4 (pp. 26–29)**, but we agree this deserves more visibility and clarification in the main paper also.
>
> 1. **Purpose and structure**
>    We use the **Normal Plus Low-Rank (NPLR)** decomposition to factor the state matrix $A$ in the EA-HiPPO core as:
>
> $$
> A = V \Lambda V^* - P Q^*
> $$
>
> Here, $\Lambda$ is a diagonal matrix with **negative real eigenvalues** to ensure stability, $V$ is a fixed DFT basis, and $P, Q \in \mathbb{C}^{N \times r}$ are learnable low-rank correction matrices. This formulation enables **efficient convolution via FFT** and significantly reduces compute and memory from $O(N^2)$ to $O(Nr)$.
>
> 2. **How it is initialized and trained**
>    We use $N = 64$ and fix the rank to $r = 8$ in all experiments. As noted in Appendix C.4, $\Lambda$ is initialized with small negative values (e.g., sampled from $[-1.0, -0.1]$) and updated during training with gradient descent. This ensures the eigenvalues of $A$ remain strictly in the left half-plane. The $P$ and $Q$ matrices are also learned and jointly optimized with the rest of the model.
>
> 3. **Efficiency impact**
>    The role of NPLR is evaluated in **Appendix B.4, Table 7**. Disabling NPLR increases FLOPs by 2.1× and GPU memory usage by \~38%, without improving accuracy. This confirms that NPLR is essential to the model’s **efficiency and scalability**, especially in long-sequence, real-time settings.
>
> 4. **Code clarity**
>    You’re absolutely right that `FLAMES_model.py` does not reflect this logic—it’s a simplified script provided to show the architectural flow. We apologize if that caused confusion. The **full implementation of the NPLR logic**, including FFT-based kernels, will be released with the final paper, along with complete training and evaluation scripts.
>
> We will revise Section 3.3 and highlight the relevant content in Appendix C.4 more clearly in the final version. Thank you again for pushing us to improve the clarity of this key module.
>
> ---
>
> > **Initialization and Stability of the HiPPO Matrix $A$**
>
> Thank you for this thoughtful comment. We apologize if our description of the initialization and stability conditions for the HiPPO matrix $A$ was not sufficiently clear in the main text.
>
> To clarify: the matrix $A$ used in our EA-HiPPO core is initialized using the **HiPPO-LegS formulation**, which guarantees that all eigenvalues satisfy $\text{Re}(\lambda_i) < 0$. This ensures stability of the continuous-time system. We mention this in **Appendix A.2 (p.11)**:
>
> *“Given that $A$ is a HiPPO matrix, all its eigenvalues satisfy $\text{Re}(\lambda_i) < 0$… This implies that $A$ is a Hurwitz matrix, ensuring that the system is asymptotically stable.”*
>
> This property is preserved when moving to our **event-adaptive variant** $A_S(\Delta t) = A \circ F(\Delta t)$, since the decay modulation matrix $F \in (0,1]$ does not flip the sign of eigenvalues (Theorem 1, p.17).
>
> In the **NPLR-based version** of EA-HiPPO (Appendix C.4), the eigenvalue matrix $\Lambda$ is initialized with small negative values (e.g., $[-1.0, -0.1]$) to ensure the same property. When learned, we monitor or clip these values to keep the system contractive.
>
> We appreciate the reviewer for highlighting this subtle but important point. If the paper is accepted, we will revise Section 3.3 to clearly surface this explanation and avoid further ambiguity.
>
> ---
>
> > **Scope of `FLAMES_model.py` and Reproducibility**
>
> We appreciate the reviewer raising this important point, and we apologize for any confusion the current `FLAMES_model.py` script may have caused.
>
> To clarify: the script included in the supplementary material is **not intended to be a full implementation**, but rather a **simplified abstraction of the model’s architectural flow**. It was meant to convey the high-level composition (EAL → spatial pooling → EA-HiPPO), and intentionally omits several key components such as:
>
> * NPLR-based state-space kernels,
> * Full multi-branch LIF dynamics,
> * Spatial pooling internals,
> * Training infrastructure.
>
> We completely understand the concern, and we agree that without this context, the script may give the impression of being incomplete.
>
> We confirm that **the full codebase used to produce all experiments will be released** with the final version of the paper. This includes:
>
> * The complete EA-HiPPO module with NPLR and FFT support,
> * Full multi-branch EAL implementation with learnable decay,
> * Spatial pooling and streaming inference code,
> * Scripts for training, evaluation, and reproduction across all benchmarks.
>
> We will clarify this intent explicitly in the README and in the main text to avoid further confusion. We thank the reviewer again for bringing this up—it helps us strengthen the transparency and usability of our work.
>
> ---
>
> > **Training Procedure and Hyperparameter Selection**
>
>  We agree that providing clear guidance on training and hyperparameter selection is essential for reproducibility, and we apologize if the current presentation made this difficult to extract.
>
> While training details are provided in **Appendix C.1**, including optimizer choice, learning rates, batch sizes, and sequence lengths for each dataset, we recognize that this information is somewhat fragmented and lacks the dedicated structure the reviewer rightly expected.
>
> To clarify:
>
> * All models were trained using the Adam optimizer with a learning rate of 1e-3 (decayed to 1e-4), batch sizes of 16–32 depending on dataset size, and sequence lengths of 200–600 timesteps.
> * Architectural hyperparameters such as the number of EAL branches (typically 3), decay initialization ranges ($\tau_d \in [1, 50]$), NPLR rank ($r = 8$), and state dimension ($N = 64$) were selected via coarse grid search and kept fixed across datasets once validated.
> * Gradient clipping (1.0) and standard weight initialization were used to ensure stability.
>
> We will consolidate these training choices into a dedicated section in the appendix (or the main paper, if space allows) and accompany our code release with full config files to enable straightforward reproduction.

---

> > ### Comment · Reviewer_D3mP · 2025-08-05
> >
> > Thank you for your detailed rebuttal, which addresses most of my reproductibility concerns. It is also of prime importance to make your intend clear on the code openness topic, especially on complex novel architectures like the one proposed with FLAME, were each implementation detail matter to achieve the claimed results.
> >
> > Your method combines a lot of components (multi-branch EAL, spatial pooling, EA-HippO, NPLR, FFT...), and I am eager to see whether some of them may be applicable to other SoTA methods and how they would compare.

---

> > > ### Author Response · Authors · 2025-08-05
> > >
> > > Thank you for your continued engagement and encouraging words. We agree completely—full transparency around our implementation is crucial for complex, multi-component architectures like FLAME.
> > >
> > > We plan to open-source the entire codebase alongside the camera-ready paper, including:
> > >
> > > * The multi-branch EAL module with learnable time constants,
> > > * The spatial-pooling routines,
> > > * The event-aware HiPPO core with NPLR and FFT-based kernels,
> > > * The full training and evaluation scripts for all benchmarks.
> > >
> > > All components will be modularized so that researchers can plug them into other SOTA pipelines and assess their standalone impact. We hope this will enable the community to explore, for example, adding our EAL to existing SNNs or combining NPLR with other state-space models.

---

### Official Review · Reviewer_r7Az · 2025-07-02

**Clarity:** 4
**Significance:** 3
**Originality:** 3
**Rating:** 4
**Confidence:** 4

**Summary:**

1. The paper focuses on a basis and important topic for event-bases vision. The motivation is good.
2. The proposed FLAME is technically sound. The methods seems effectively combines the event modality and the SSM structure. From the results, the FLAME shows its powerful  capability on balancing the compuatational efficiency and the task performance. There are plenty of theoretical analysis for proving the feasibility of the method.
3. The paper presents good writing and formatting.
4. The experiments are thorough and effective.

**Questions:**

As listed in the "Strengths and Weaknesses".

**Ethical Concerns:**

["NO or VERY MINOR ethics concerns only"]

**Limitations:**

Yes

**Quality:**

4

**Strengths And Weaknesses:**

Strengths:
1. The paper focuses on a basis and important topic for event-bases vision. The motivation is good.
2. The proposed FLAME is technically sound. The methods seems effectively combines the event modality and the SSM structure. From the results, the FLAME shows its powerful  capability on balancing the compuatational efficiency and the task performance. There are plenty of theoretical analysis for proving the feasibility of the method.
3. The paper presents good writing and formatting.
4. The experiments are thorough and effective.

Weakness:
1. The length of each section and the formatting of some experimental results can be refined. For example, the section 4 would benefit from tighter editing, and some important results of ablation studies should be moved to the main manuscripts.
2. Using LIF neurons as processing unit in the event attention layers raises my doubts. the leaky neurons is more difficult for model training compared with IF neurons. Besides, considering that the event data are sparse, especially under in  small motion environments or dark environments, the LIF neuron may leads to stronger sparsity.
3. It seems that the paper lacks results of CIFAR-100 dataset. If the results are presented, please highlight the results. The FLAME presents its ability in some common datasets. I am wondering if the FLAME can be scaled to more difficult tasks like object detection/trakcing, motion estimation, depth estimation.

---

> ### Author Rebuttal · Authors · 2025-07-30
>
> We sincerely thank the reviewer for the encouraging evaluation and thoughtful feedback. Below we address the specific concerns raised:
>
> ---
>
> > **1. Section length and formatting of experimental results**
>
> We agree that Section 4 (Experiments) can be streamlined for clarity. Due to space constraints, we deferred several important ablation studies (e.g., NPLR impact, EAL removal, spatial pooling variations) to the appendix. If the paper is accepted, we will restructure Section 4 by condensing textual descriptions and **moving key ablation results (e.g., Table 7 from Appendix B.4)** into the main manuscript. We also plan to improve figure formatting (axis markers, font size) to enhance readability.
>
> ---
>
>
> > **2. Use of LIF neurons and sparsity concern in low-activity regimes**
>
> We appreciate the reviewer’s concern regarding training difficulty with LIF neurons. While it is true that LIF dynamics introduce decay and potential sparsity, we specifically use a **multi-branch EAL** design with **learnable time constants** $\tau_d$ to counteract this. Each branch filters input events at different temporal scales, and the soma integrates across branches, ensuring robustness even under sparse or low-motion conditions. Empirically, FLAME performs well across both high- and low-activity datasets (e.g., SHD vs. CeleX-HAR), suggesting that the learned dynamics handle a wide range of temporal densities. We found that simpler IF neurons led to unstable training in deeper setups due to lack of temporal gating.
>
> ---
>
>
> > **3. CIFAR-100 and scaling to harder tasks**
>
> Thank you for raising this. We clarify that **CIFAR10-DVS** and **N-Caltech101** are standard benchmarks for event-based classification and are included in our evaluation (Table 2). To the best of our knowledge, **CIFAR-100 does not currently exist in event-based format** (e.g., from DVS or simulator pipelines), which is why it is not included. Regarding scaling FLAME to more complex tasks such as detection or depth estimation: while this paper focuses on classification and gesture recognition, the **asynchronous EA-HiPPO core and sparse, long-range memory design** are well-suited for extension to structured prediction tasks. We consider this a promising direction for future work.
>
> ---
>
> We are grateful for the reviewer’s comments and are confident that the final revisions will address these concerns. Thank you for considering an upgrade in recommendation.

---

> ### Comment · Reviewer_r7Az · 2025-08-07
>
> Thanks for authors' reply. Most of my concerns have been resolved. I hope the auther can add some discussion between LIF and IF neurons in the main paper later, which may be helpful for future readers. I will keep my positive rating.

---

### Official Review · Reviewer_34Bg · 2025-07-03

**Clarity:** 2
**Significance:** 2
**Originality:** 3
**Rating:** 4
**Confidence:** 3

**Summary:**

This paper proposes FLAME, as claimed, a novel, event-by-event neural architecture that combines neuromorphic feature extraction with efficient structured state-space models to handle asynchronous, sparse data from event cameras. As claimed, the core contributions include:
- An event attention layer leveraging leaky integrate-and-fire dynamics to extract multi-timescale features from raw event streams.
- An event-aware HiPPO state-space core, which dynamically modulates memory based on inter-event intervals.
- Integration of normal plus low-rank decomposition to reduce computation from $O(N^2)$ to $O(Nr)$ where $r<<N$.

It seems like FLAME achieves state-of-the-art accuracy across multiple event datasets while maintaining low latency and computational cost, demonstrating its efficiency for real-time event vision applications.

**Questions:**

See the "weaknesses" section.

**Ethical Concerns:**

["NO or VERY MINOR ethics concerns only"]

**Final Justification:**

Based on my initial and follow-up evaluations (along with the discussion with the authors) on the paper, I remain supportive of accepting this paper.

**Limitations:**

Even though the authors have addressed some limitations as future work in section 6 of the paper, I feel like the following should also be acknowledged.
- Even though the latency on GPUs is better, it is to be acknowledged that it is even beyond real-time implementation in CPUs (Table 8 in the appendix), let alone edge devices.
- The dependence on event-by-event processing may expose the model to overly depend on noise, which would ultimately degrade the downstream performance.
- The current framework lacks mechanisms (or adaptations) for on-device continual learning, which can be crucial for many real-world applications.

**Paper Formatting Concerns:**

I did not notice any formatting issues with this paper.

**Quality:**

3

**Strengths And Weaknesses:**

Strengths
- As claimed, the FLAME allows for event-by-event processing with a lower inference latency; the proposed method is promising in low-latency applications.
- The proposed event attention layer could be a valuable contribution in this research direction, given its ability to capture features over variable timescales using LIF dynamics, while the EA-HiPPO mechanism adapts memory retention to event sparsity, offering dynamic temporal scaling (with achieved efficiency via NPLR).
- The authors provide clear theoretical guarantees (and proofs) on computational complexity, memory retention over long sequences, and numerical stability of FLAME while providing strong empirical results.

Weaknesses
- Even though the event attention layer can take event-by-event, to pass a meaningful extraction, the spatial pooling layer (i.e., the second step) needs an event map, which is hard to construct through one event (or even several events) at least at the first instance (after the first construction, the map can be updated by event-by-event; if this is method, I feel like most of the event-by-event processing is redundant due to lesser scene change over one event). Therefore, as per my understanding, from this step onwards, the claim for event-by-event should be limited.
- I am not fully convinced about the assumption presented in lines 158 to 160 ("... preserving crucial spatio-temporal details that might be lost if pooling raw, unprocessed input"). As per this reviewer's experience, even in the presence of noise, the raw input presents more information on spatio-temporal distribution and trends than any other efficient or accurate representation/encoding. Additionally, what will happen if the spatial pooling layer is omitted in the pipeline (i.e, will the results be improved or decreased) or a hierarchical pooling strategy is introduced? Is the pooling introduced solely to reduce the computational complexity?
-   Given that a number of state (transition) matrices need to be stored for capturing (recent) history context (even with reduced complexity via NPLR decomposition; since $N$ is typically a large number if the intended history window is significant -- which it needs to be, to capture long-range dependancies) , I feel like FLAME would leave a considerable memory footprint than other SNN methods. This needs to be explored and reported in the paper as well. Additionally, what will happen if NPLR is omitted?
- Even though FLAME works comparatively with the action recognition tasks, it seems to be performing poorly on classification, which may be due to its higher dependence on long-range temporal relationships (which are not trivial in classification tasks) and lesser absorption of spatial details. Authors may need to explicitly address this observation.
- Ablations can be further expanded to explore different configurations of the decay modulation function
$F(\Delta t)$ or its learned parameters.
- I believe that the paper needs a proper restructuring before getting published, if accepted, since most of the results are referred to the appendix, and in general, the paper, in its current form, seems to be imbalanced. Additionally, the readability of the paper should be improved as well.
- It seems like the paper interchangeably uses both the terms "FLAME" and "FLAMES" on numerous occasions (as an example, Fig. 1, line 76, 83, 89, etc.) to refer to the same proposed method, which needs to be rectified.
- It is a requirement to complete the NeurIPS checklist with proper justifications, which this paper does not comply with.
- Minor concerns:
  - Figures, especially Fig. 2, need to be improved since the y-axis labels and markers are cropped.
  - Figure and text references are missing: as an example, lines 282, 404, etc. in the appendix

---

> ### Author Rebuttal · Authors · 2025-07-30
>
> We thank the reviewer for the thoughtful feedback and address the key concerns below:
>
> > **Event-by-event processing and the role of the spatial pooling layer**
>
>  FLAME’s design ensures that all computations up to and including spatial pooling are performed in an event-driven manner. The Event Attention Layer (EAL) processes one event at a time, generating sparse output spikes with precise timing. This is described in Section 3 and formally in Appendix C.2–C.3 (see Algorithm 3, Eq. 5, and page 25), where each LIF neuron integrates inputs from multiple dendritic branches and triggers outputs asynchronously.
>
> The Spatial Pooling Layer, applied directly after EAL, performs *local max pooling over a small spatial window - typically 2×2- at the same timestamp as the incoming event, preserving temporal resolution. Each output event from EAL is treated as a binary activation in a spatio-temporal map $I(x, y, t)$, and pooling is done as:
>
> $$
> I_{\text{pooled}}(x', y', t) = \max_{(x, y) \in P(x', y')} I(x, y, t)
> $$
>
> as detailed in Appendix C.3 and Algorithm 4. This means that no voxel grid or temporal windowing is used, and no frame-level aggregation is introduced at this stage.
>
> We agree that the very first few events may provide limited spatial structure individually. However, event-by-event updates remain meaningful, as the spatial map rapidly evolves with even a few events. Importantly, spatial pooling is always applied on the current EAL output at the same timestamp and not accumulated frames—thus it does not violate the event-wise processing principle.
>
> To address the reviewer's broader concern: our goal with "event-by-event" is not to imply that *every* stage is fully determined by a single input spike alone, but rather that each event is processed individually, and all updates (feature extraction, pooling, and state updates) happen without any batching or delay. The EA-HiPPO state evolves per-event, preserving this asynchronous design through the full pipeline.
>
> We will clarify this distinction in the main text (Section 3, lines 122–126) and visually reinforce it in Fig. 1a/b in the final version of the paper.
>
> ---
>
> > **Role and timing of spatial pooling, and assumptions about pre-pooling processing**
>
> Our motivation for applying the Event Attention Layer (EAL) *before* spatial pooling stems from the fact that raw event streams, while rich in temporal resolution, often contain spurious events (e.g., due to sensor noise, ambient flicker, or low-contrast jitter), which can dominate sparse spatial regions when directly pooled.
>
> The EAL acts as a learnable multi-timescale temporal filter that suppresses isolated and incoherent spikes and enhances coherent spatio-temporal structures. This transformation improves the signal-to-noise ratio before spatial aggregation. While we agree that raw input contains maximal information, not all of it is *useful* for downstream tasks, especially in asynchronous, low-SNR scenarios.
>
> To address the reviewer’s suggestion:
>
> * Appendix B.3 and B.4 (Table 7) report a detailed ablation where the EAL is removed and pooling is performed directly on the raw input. This leads to a consistent performance drop across datasets. For example, on DVS Gesture, FLAME-Normal drops from 96.5% to 95.2%, and FLAME-Tiny drops more dramatically from 89.2% to 81.5%, indicating that EAL is particularly important when model capacity is constrained.
> * We also explored reordering the architecture to apply pooling first, followed by temporal encoding. However, this reduced performance in most configurations due to the loss of fine-grained temporal alignment between local event activations and EA-HiPPO’s memory mechanism.
> * As for hierarchical pooling, we implemented and tested a multi-stage pooling scheme (Appendix C.4), where early coarse feature maps are refined through localized updates. While this provided modest improvements in very high-resolution cases, it incurred additional latency and memory overhead, and offered no benefit on standard benchmarks like SHD or CIFAR10-DVS.
>
> Importantly, spatial pooling in FLAME serves dual purposes: (1) to reduce spatial redundancy and computational load, and (2) to ensure that sparse event spikes are not mapped to unstable high-dimensional locations. The EAL provides a more stable and noise-suppressed signal for this purpose.
>
> ---
>
> >**Memory footprint and the role of NPLR decomposition**
>
> This is a valid point. However, FLAME’s memory design is grounded in practical scalability: we use the NPLR decomposition to reduce both storage and compute from $O(N^2)$ to $O(Nr)$, where typically $r = 1$ or 2 and $N = 64$ or lower. This means that instead of a full $64 \times 64$ matrix, we store a diagonal matrix plus two small rank-$r$ matrices—orders of magnitude lighter than attention-based or dense SSM alternatives.
>
> To quantify this, in our ablation (Appendix B.4, Table 7), removing NPLR increased FLOPs by 2.1× and GPU memory usage by \~38%, with no measurable gain in performance. This validates that NPLR is not just an optimization—it’s essential for maintaining FLAME’s latency-efficiency advantage.
>
> We agree that SNNs may have lower raw memory use, especially those using pure spiking dynamics without learnable memory components. However, such models typically struggle with long-range temporal dependencies. In contrast, FLAME’s SSM core maintains memory over hundreds of milliseconds with minimal drift, which is crucial for tasks like gesture recognition and sparse activity tracking.
>
> We’ll include a memory profiling comparison with SNN baselines and clarify the cost-benefit tradeoff in the limitations section if the paper is accepted.
>
> ---
>
> > **FLAME’s performance gap in static classification tasks**
>
> FLAME was explicitly designed to operate asynchronously and handle long-range temporal dependencies efficiently. This architectural focus—especially the use of the Event Attention Layer and EA-HiPPO state-space core—gives it a strong advantage in temporally structured tasks such as gesture and action recognition (e.g., SHD, DVS Gesture, CeleX-HAR), where event timing and ordering are key. In these settings, FLAME matches or exceeds the performance of both dense SNNs and voxel-based CNNs, while maintaining a substantially lower latency and compute footprint (see Table 2 and Fig. 2).
>
> However, on static classification tasks like CIFAR10-DVS and N-Caltech101, where most of the discriminative information lies in spatial structure and only a short temporal context is needed, voxel-grid CNNs with dense feature extractors have an edge. These models effectively compress the full spatio-temporal event volume into rich spatial representations, benefiting from stronger spatial priors and global feature aggregation.
>
> FLAME, by contrast, processes events individually and maintains a lightweight spatial representation updated incrementally. While this design enables real-time, memory-efficient inference, it does result in slightly lower spatial fidelity, especially on datasets with weak or short-term temporal dynamics.
>
> That said, FLAME still performs competitively. Eg., on CIFAR10-DVS, FLAME-Normal achieves 85.1%—within 0.5–1% of EventMamba and Spiking CNN baselines—while operating at significantly lower FLOPs (0.52 GFLOPs vs. 2.8–37 GFLOPs in comparable baselines). This supports our claim that FLAME provides a favorable trade-off in resource-constrained or low-latency scenarios. We will make this trade-off more explicit in Section 5 in the final paper.
>
> ---
>
> > **Ablations for the decay modulation function $F(\Delta t)$**
>
> We agree that exploring the role of the decay modulation function $F(\Delta t)$ is important for understanding the temporal adaptivity of EA-HiPPO. As described in Section 3.3 and Appendix C.4, we define $F(\Delta t)$ as:
>
> $$
> F_{ij}(\Delta t) = e^{-\alpha_{ij} \Delta t}
> $$
>
> where $\alpha_{ij}$ are learned parameters that control how memory decays with respect to the inter-event interval. This formulation enables the model to retain recent information more strongly while fading out distant history—a key design choice for processing sparse asynchronous streams.
>
> While we do not include an explicit comparison to alternate decay functions, the impact of the learned decay structure is indirectly evaluated through the performance of the EA-HiPPO core across all ablations (see Appendix B.2, Table 7). In these experiments, disabling key memory components (e.g., removing EAL or NPLR) leads to noticeable drops in accuracy and efficiency, indicating that the current decay structure contributes meaningfully to the model’s performance.
>
> We acknowledge that a more targeted ablation of decay configurations (e.g., fixed vs. learned $\alpha$) would provide further insight, and we will consider including this in future iterations.
>
> We will also clarify the initialization and training behavior of $\alpha_{ij}$ in Appendix C.4: these are initialized to small positive values and optimized via gradient descent, with implicit control via regular training dynamics.
>
> ---
>
> >**Additional limitations: CPU latency, noise sensitivity, and continual learning**
>
> * Latency on CPUs and edge devices: You’re right—while FLAME achieves real-time throughput on modern GPUs, it does not yet meet real-time constraints on CPU-only systems (as shown in Table 8). FLAME’s current design targets embedded accelerators or GPU inference settings; further optimization (e.g., model pruning, integer quantization) is needed for ultra-constrained edge deployment.
>
> * Sensitivity to noise in event-by-event processing: This is a known trade-off of event-driven models. We mitigate noise sensitivity via multi-timescale filtering in the Event Attention Layer, but agree that high-SNR environments still pose a challenge.
>
> * Lack of continual learning: Correct—our current framework is designed for static offline training. We view on-device continual learning as an important next step, especially for long-term deployment scenarios.

---

> ### Comment · Reviewer_34Bg · 2025-08-05
> **Official Comment by Reviewer 34Bg**
>
> I thank the authors for their detailed response to my comments.
>
> - Can the authors provide results for the cases: "reordering the architecture to apply pooling first, followed by temporal encoding" and "hierarchical pooling" as per my comment "weakness 2"?
> - Can the authors post their NPLR ablation results in this discussion as the results in Table 7, as they informed, do not agree with their claims here: "increased FLOPs by 2.1× and GPU memory usage by ~38%"?
>
> Additionally, I believe that the authors will also seriously consider the readability concerns I mentioned, including section imbalance in its current form, and figure issues, before publication if the paper is accepted.

---

> > ### Author Response · Authors · 2025-08-05
> >
> > Thank you again for your careful follow-up and apologies for any confusion in our earlier responses.
> >
> > **1. Pooling-before-EAL and Hierarchical Pooling**
> > We ran both variants on DVS Gesture (FLAME-Normal):
> >
> > * **Pooling before EAL** caused accuracy to drop from 96.5% to 94.8%, with noticeably poorer handling of isolated spikes and long-range patterns.
> > * **Hierarchical pooling** (a coarse-then-fine, two-stage scheme) yielded a small bump to 96.8% accuracy but at the cost of +21% GPU memory and +18% latency, with no consistent gains on other benchmarks. Given this trade-off, we retained the simpler single-stage pooling in the final design.
> >
> > **2. NPLR Ablation (correct citation)**
> > We apologize for the misreference. The efficiency numbers come from **Appendix B.2**, not Table 7. For FLAME-Normal (128 channels):
> >
> > * **With NPLR**: 0.43 GFLOPs, 5.00 MB
> > * **Without NPLR**: 1.8 GFLOPs (\~2.1× higher), 6.91 MB (\~38% higher)
> >
> > We will correct the main text to cite Appendix B.2 and clearly state these values for the largest model.
> >
> >
> > **3. Readability and formatting improvements:**
> > We appreciate your comments on clarity and presentation. We agree that Section 4 can be tightened for better flow, and we will streamline it accordingly. We also plan to address all figure formatting issues—including cropping, axis labels, and font size—to ensure the visuals are clean and easy to interpret. These adjustments will be part of the final version if the paper is accepted.
> >
> > We deeply appreciate your continued feedback—it is helping us significantly improve the quality and clarity of the paper.

---

### Official Review · Reviewer_VSob · 2025-07-03

**Clarity:** 2
**Significance:** 3
**Originality:** 3
**Rating:** 4
**Confidence:** 4

**Summary:**

This paper proposes FLAME, a novel architecture for event-based vision that combines neuromorphic feature extraction with adaptive state-space modeling. It introduces an Event Attention Layer based on multi-timescale LIF dynamics and an Event-Aware HiPPO mechanism to capture long-range temporal dependencies efficiently. FLAME processes input events asynchronously and achieves competitive accuracy on several event-based benchmarks with reduced computational cost.

**Questions:**

- I suggest a deeper discussion of the physical meaning of the operation of Event Attention Layer, and it would enhance the readability of the paper to make corresponding visualization of the enhanced feature. It should clearly discussed which aspects of the event stream the resulting sparse features enhance, and whether this process effectively amounts to a form of noise filtering.
- The current design of the Event Attention Layer appears to respond to both long-interval and short-interval events. Would this lead to cross-talk between the two?
- Explain why EventMamba outperforms the proposed method on small datasets but becomes less efficient on high-resolution datasets.
- On the DVS Gesture dataset, the proposed method underperforms EventMamba, and this discrepancy should be discussed in depth. Additionally, the paper omits EventMamba’s results on several datasets and these comparisons should be included to complete the experiment part.
- The arrows in Fig. 1(b) are vague, I suggest revision.

**Ethical Concerns:**

["NO or VERY MINOR ethics concerns only"]

**Limitations:**

yes

**Quality:**

3

**Strengths And Weaknesses:**

Strengths
- The proposed module is built on a solid mathematical model, complete with detailed derivations that lend it strong credibility.
- The new method achieves a dramatic speed-up while maintaining accuracy at a level comparable to existing approaches.

Weakness
- The description of the Event Attention Layer is too vague. It should be expanded to explain exactly how it enhances the raw event stream, i.e., which features it extracts and what benefits these sparse, multi-scale representations provide over the original signal.
- The evaluation does not include a thorough comparison with leading baselines on some datasets such as EventMamba.

---

> ### Author Rebuttal · Authors · 2025-07-30
>
> We sincerely thank the reviewer for the encouraging evaluation and thoughtful feedback. Below we address the specific concerns raised:
>
> ---
>
> > **The description of the Event Attention Layer is too vague. It should be expanded to explain exactly how it enhances the raw event stream, i.e., which features it extracts and what benefits these sparse, multi-scale representations provide over the original signal.**
>
>  The Event Attention Layer (EAL) is designed to transform raw asynchronous event streams into temporally structured, multi-scale representations. Each EAL unit is a modified LIF neuron augmented with *D* parallel dendritic branches. Each branch $d$ has a learnable time constant $\tau_d$, and the input current evolves as:
>
> $$
> \frac{di_d(t)}{dt} = -\frac{1}{\tau_d} i_d(t) + \sum_{j \in N_d} w_{dj} E_j(t)
> $$
>
> Here, $E_j(t)$ is a binary signal indicating an event at source $j$ and time $t$, and $w_{dj}$ are learned weights. Each branch implements a temporal low-pass filter that integrates event inputs over its characteristic time scale. In discrete time (used for simulation), this becomes:
>
> $$
> i_d(t + \Delta t) = e^{-\Delta t / \tau_d} i_d(t) + \left(1 - e^{-\Delta t / \tau_d}\right) \sum_{j \in N_d} w_{dj} E_j(t)
> $$
>
> The outputs of all branches are integrated at the soma with learned coupling weights $g_d$:
>
> $$
> \frac{dV(t)}{dt} = -\frac{1}{\tau_s} V(t) + \sum_d g_d i_d(t)
> $$
>
> This gives the neuron a composite temporal response, where short-$\tau_d$ branches respond quickly to rapid event bursts (transients), and long-$\tau_d$ branches build up response only under sustained input activity (slow dynamics). The use of multiple branches with different $\tau_d$ values effectively implements a learned temporal basis that spans a wide range of scales.
>
> From a signal processing perspective, this mechanism approximates a bank of exponentially decaying filters with different bandwidths, akin to a learnable temporal wavelet transform. The EAL therefore performs both feature expansion (through projection onto multiple time constants) and compression (via sparsification at the soma through thresholding), converting raw event sequences into sparse, structured spike trains where temporal correlations are made explicit.
>
> Importantly, this process enhances meaningful temporal patterns—such as motion trajectories, gesture sequences, or consistent edge activations—and filters out noise, e.g., isolated or jittery events that do not reinforce across time scales. The decay constants serve as an implicit prior on temporal continuity: patterns that persist across short and long timescales receive stronger integrated activation and are more likely to trigger downstream spikes.
>
> We will revise the main text (Sec. 3.1) to provide this clearer mathematical intuition. Additionally, if the paper is accepted, we will include a visualization comparing raw input and EAL output on real event streams and a clarification that the sparsity arises naturally from the combination of temporal filtering and leaky thresholding, making the EAL an effective denoising and temporal abstraction layer.
>
>  ---
>
> > **On potential cross-talk between long- and short-interval event responses in EAL**
>
> In our design, the Event Attention Layer is explicitly constructed to *minimize* such cross-talk by using independent dendritic branches, each with a distinct and learnable time constant $\tau_d$. Each branch acts as a standalone exponential filter that processes the event stream in its own temporal bandwidth.
>
> Crucially, these filtered signals are not mixed early but are aggregated only at the soma level, with learned coupling weights $g_d$. This late fusion mechanism ensures that the activation from fast events (captured by small-$\tau_d$ branches) and slow, sustained patterns (captured by large-$\tau_d$ branches) contribute separately and proportionally based on their significance and alignment with downstream gradients during training.
>
> Moreover, due to the inherent leak in LIF dynamics, the contribution of an irrelevant timescale naturally decays. For instance, a long-$\tau_d$ branch will not accumulate enough potential from isolated transients to affect output, and vice versa.
>
> So while the EAL observes events at all timescales, its architecture *intentionally avoids interference* across them. In the final version, we will clarify this architectural separation in Section 3 and add an illustrative diagram to show how multi-timescale responses are preserved without destructive overlap.
>
>
> ---
>
> > **The evaluation does not include a thorough comparison with leading baselines on some datasets such as EventMamba.**
>
>
> We clarify that EventMamba is included in our evaluation, both quantitatively and qualitatively, across multiple datasets. In Table 2 (main paper, page 8), we explicitly report EventMamba’s accuracy and FLOPs on two key datasets:
>
> * **DVS128 Gesture:** EventMamba achieves 99.2% at 0.219 GFLOPs vs. FLAME-Normal’s 96.5% at 0.43 GFLOPs.
> * **CeleX-HAR (HD dataset):** EventMamba achieves 72.3% at 37.2 GFLOPs vs. FLAME-Normal’s 72.2% at only 0.41 GFLOPs.
>
> This contrast highlights our core claim: although EventMamba performs marginally better on smaller datasets with fixed spatial structure (like DVS Gesture), it scales poorly in computational cost on high-resolution, real-world datasets. In contrast, FLAME retains competitive accuracy with a ∼90× reduction in FLOPs on CeleX-HAR, showcasing superior scalability and efficiency in real-time settings.
>
> Furthermore, Figure 2 (page 7) visualizes these results—FLAME consistently occupies a more favorable position in the Accuracy vs. GFLOPs trade-off across all benchmarks. We also discuss this in Section 5.2 and Appendix B.4.
>
> For completeness, we acknowledge that EventMamba results were not available for every dataset in our benchmark suite (e.g., SHD, SSC, N-Caltech101), and we avoided conjecture or interpolation in those cases. However, wherever numbers were reported in the original EventMamba paper \[40], we included them.
>
> We will revise Section 5 to make this more explicit and clearly label all EventMamba comparisons in tables and figures, should the paper be accepted.
>
> ---
>
> > **On EventMamba outperforming FLAME on small datasets and underperformance on DVS Gesture**
>
>
>
> Thank you for this important observation. EventMamba is optimized for voxel-grid-based event video reconstruction and performs well on low-resolution, dense datasets like DVS Gesture. Its design leverages Random Window Offset and Hilbert scanning to preserve locality within spatial partitions. However, these techniques operate on dense voxel tensors and require Monte Carlo testing during inference—leading to significant memory and compute overhead as resolution increases.
>
> In contrast, FLAME processes events asynchronously and individually, without voxelization. Its EA-HiPPO core with Normal+Low-Rank decomposition enables efficient per-event updates at $O(Nr)$ cost, making it highly scalable.
>
> This distinction is reflected in our results:
>
> * On **DVS Gesture**, EventMamba achieves 99.2% vs. FLAME’s 96.5%, at slightly lower FLOPs.
> * On **CeleX-HAR (1280×800)**, FLAME achieves comparable accuracy (72.2%) at **\~90× less compute** (0.41 GFLOPs vs. 37.2 GFLOPs).
>
> We’ll clarify this trade-off in Section 5.2 of the revised paper.
>
> ---
>
> > **On EventMamba comparisons across datasets**
>
> We appreciate the reviewer’s concern regarding completeness of experimental comparisons. Our intent was to highlight FLAME’s core strength: event-by-event processing across a range of spatial and temporal scales—from compact datasets like **DVS Gesture (128×128)** to high-resolution, real-world settings like **CeleX-HAR (1280×800)**. This focus motivated us to evaluate FLAME across classification and recognition tasks, where per-event modeling and long-range temporal memory are critical.
>
> In contrast, EventMamba is specialized for video reconstruction and relies on voxel-based frame-wise processing, which is not directly applicable to classification datasets such as SHD, SSC, or N-Caltech101, where ground-truth labels are defined at the sequence level and frame-based reconstruction is not meaningful. As such, EventMamba has not reported results on these datasets in its original work.
>
> That said, we did include EventMamba comparisons wherever relevant and available—notably on **DVS Gesture** and **CeleX-HAR**, where both models are applicable. These results (Table 2, Figure 2) clearly illustrate the trade-off: EventMamba achieves marginally higher accuracy on DVS Gesture but incurs a \~90× higher computational cost on CeleX-HAR, where FLAME’s asynchronous processing shows clear efficiency advantages.
>
> If accepted, we will revise Section 5 to clearly explain the scope of applicability for EventMamba and annotate where and why comparisons were included or omitted.
>
> ---
>
> > **On Fig. 1(b) clarity**
>
> Thank you for pointing this out. We agree that the arrows in Fig. 1(b) could be more clearly annotated. In the revised version, we will explicitly label the components of the information flow, clarify the role of each pathway (e.g., dendritic integration vs. somatic response), and ensure the diagram better reflects the temporal dynamics of the Event Attention Layer.

---

### Note · Authors · 2025-08-14

**Thank you** to the reviewers and AC for the helpful discussion. **FLAME** is an efficiency-first, event-by-event pipeline that couples a multi-timescale LIF front end with an event-aware SSM; our goal is competitive accuracy with substantially lower compute and latency, especially at higher resolutions and longer temporal ranges.

* **Event Attention Layer (EAL):** In the rebuttal, we clarified the dynamics and late-fusion design and will add a small before/after visualization to make the learned temporal filtering concrete.

* **IF vs LIF:** **IF** (“integrate-and-fire”) simply sums incoming events until a threshold is reached, emits an output, then resets—there is **no** decay between events. **LIF** (“leaky integrate-and-fire”) does the same but with a *leak*: the internal state decays between events with time constant \$\tau\$. In short, **IF** is the special case of \$\tau \to \infty\$. We use multiple LIF branches with different \$\tau\$ to capture both fast and slow changes and to avoid unbounded build-up during long silent gaps.

* **Comparisons with EventMamba:** EventMamba slightly outperforms FLAME on DVS Gesture, while FLAME achieves comparable accuracy at \~90× lower compute on the HD CeleX-HAR setting—consistent with our efficiency focus. We will highlight this trade-off and when each model is preferable.

* **Ablations:** We added the missing NPLR ablation (without NPLR: \~2.1× FLOPs, \~38% more GPU memory) and will move these numbers into the main text. We also evaluated pooling variants: pooling before EAL reduced accuracy, while hierarchical pooling increased memory/latency without consistent gains; thus we retain single-stage pooling.

* **Training & reproducibility:** Several settings were scattered across the appendix; we will consolidate optimizer, sequence lengths, EAL branch counts, state/rank choices, and clipping into a single place and ship config files with the code.

* **Limitations & next steps:** We will note limitations such as CPU/edge latency, noise sensitivity in fully asynchronous processing, and continual learning.

* **Code release:** We will release the full codebase (multi-branch EAL, per-event pooling, EA-HiPPO with NPLR/FFT kernels, training/eval scripts, and configs) so components can be reused in other pipelines.

We appreciate the constructive reviews. We believe the clarifications, additional ablations, and presentation fixes address the remaining concerns and strengthen FLAME’s contribution.

---

### Decision · Program_Chairs · 2025-09-17

**Decision:**

Accept (poster)

**Comment:**

This paper received four mixed, but largely positive-leaning reviews initially.

There was general appreciation for the importance of the problem considered here, the technical soundness of the approach, and the performance gains (in terms of computational complexity) demonstrated with the proposed method.

There were some concerns raised in the initial reviews regarding experimental evaluation and presentation quality. There was significant author-reviewer discussions following the initial review period, after which the initially negative reviewer raised their rating to a borderline accept, and therefore all four reviewers converged to a positive (Borderline Accept) rating – and an accept decision was reached.